# Noninvasive quantification of axon radii using diffusion MRI

**Jelle Veraart[1,2,3]\*, Daniel Nunes[1], Umesh Rudrapatna[4], Els Fieremans[2], Derek K Jones[4,5], Dmitry S Novikov[2†], Noam Shemesh[1†]**

[1]Champalimaud Research, Champalimaud Centre for the Unknown, Lisbon, Portugal; [2]Center for Biomedical Imaging, Department of Radiology, New York University School of Medicine, New York, United States; [3]imec-Vision Lab, Department of Physics, University of Antwerp, Antwerp, Belgium; [4]CUBRIC, School of Psychology, Cardiff University, Cardiff, United Kingdom; [5]Mary MacKillop Institute for Health Research, Australian Catholic University, Melbourne, Australia

**Abstract** Axon caliber plays a crucial role in determining conduction velocity and, consequently, in the timing and synchronization of neural activation. Noninvasive measurement of axon radii could have significant impact on the understanding of healthy and diseased neural processes. Until now, accurate axon radius mapping has eluded in vivo neuroimaging, mainly due to a lack of sensitivity of the MRI signal to micron-sized axons. Here, we show how – when confounding factors such as extra-axonal water and axonal orientation dispersion are eliminated – heavily diffusion-weighted MRI signals become sensitive to axon radii. However, diffusion MRI is only capable of estimating a single metric, the effective radius, representing the entire axon radius distribution within a voxel that emphasizes the larger axons. Our findings, both in rodents and humans, enable noninvasive mapping of critical information on axon radii, as well as resolve the long-standing debate on whether axon radii can be quantified.

**\*For correspondence:**
jelle.veraart@nyulangone.org

[†]These authors contributed equally to this work

**Competing interests:** The authors declare that no competing interests exist.

## Introduction

Axons facilitate connectivity between distant neurons. Along with myelination, the axon radius determines the conduction velocity, thereby shaping the timing of neuronal computations and communication (**Waxman, 1980**). Using a model of action potential neurophysiology (**Rushton, 1951**), it has been shown that the axon radius explains the largest proportion of variance in conduction velocity (**Drakesmith et al., 2019**). Histological studies demonstrated that axon sizes vary widely within the human brain, ranging from 0.1 $\mu$m to more than 3 $\mu$m (**Aboitiz et al., 1992**; **Innocenti et al., 2015**; **Liewald et al., 2014**), and across species (**Olivares et al., 2001**; **Schüz and Preiβl, 1996**; **Liewald et al., 2014**). Moreover, axon radii have been shown to be altered in various disease processes. For example, direct axon counting in post-mortem tissue has suggested that smaller axons may be preferentially susceptible to axonal injury in multiple sclerosis (**Evangelou et al., 2001**) due to inflammation (**Campbell et al., 2014**). Electron microscopy has revealed a higher percentage of small-radius axons and a lower percentage of large-radius axons in several anatomically and functionally distinct segments of the corpus callosum in autistic subjects compared to healthy controls (**Wegiel et al., 2018**). From the animal literature, morphometric analysis of adult rat brains showed reduced axonal radii without axonal loss after chronic alcohol feeding (**Kjellström and Conradi, 1993**). Such studies indicate that non-invasive metrics capable of reporting on features of the axon radius distribution could provide important neuroimaging biomarkers for basic research and clinical applications.

A particularly relevant neuroimaging modality attuned to the microarchitecture of living brain tissue is diffusion-weighted MRI (dMRI). dMRI is sensitive to the thermal motion of water molecules

and their interference with microscopic boundaries, such as imparted by cells and subcellular structures in the brain (*Tanner, 1979*; *Le Bihan, 2003*; *Le Bihan et al., 1986*; *Callaghan et al., 1988*; *Basser et al., 1994*; *Jones, 2010*; *Beaulieu, 2002*; *Novikov et al., 2019*). Applications of dMRI specialize in revealing macroscopic brain connections (*Jbabdi et al., 2015*) and in the interpretation of contrast differences in diffusion-weighted images (*Moseley et al., 1990*; *Baron et al., 2015*). However, reproducible and specific biomarkers for studying disease onset and progression non-invasively and quantitatively in the entire brain, in particular vis-a-vis axonal properties, would confer clear advantages. Several studies have used various methods to report on axon radius parameters; still, despite many attempts, axon radius mapping using dMRI remains highly contested (*Assaf et al., 2013*; *Horowitz et al., 2015*; *Alexander et al., 2010*; *Innocenti et al., 2015*; *Xu et al., 2014*; *Burcaw et al., 2015*; *Ong et al., 2008*; *Ong and Wehrli, 2010*). Discrepancies between histology and dMRI-derived axon radii uncovered various confounding factors, for example orientation dispersion (*Drobnjak et al., 2016*; *Nilsson et al., 2012*), time-dependent extra-axonal diffusion overshadowing the intra-axonal signal at low diffusion weighting (*Burcaw et al., 2015*; *Fieremans et al., 2016*; *Lee et al., 2018*), weak signal attenuation for typically very narrow axons, especially in the realistic experimental regime of long diffusion gradient duration (*van Gelderen et al., 1994*; *Neuman, 1974*), and/or putative shrinkage during tissue preparation (*Barazany et al., 2009*; *Innocenti et al., 2015*; *Aboitiz et al., 1992*).

Recent advances in biophysical modeling and hardware prompted a revival of MR axon radius mapping (*McNab et al., 2013*; *Huang et al., 2015*; *Jones et al., 2018*). First, several of the most crucial confounding factors have been removed using powder-averaging concepts (*Callaghan et al., 1979*; *Jespersen et al., 2013*; *Kaden et al., 2016*). Averaging diffusion-weighted signals that are isotropically distributed on a sphere with constant diffusion-weighting strength $b$ has been shown to factor out the orientation dispersion (*Jespersen et al., 2013*; *Kaden et al., 2016*; *Mollink et al., 2017*), thereby eliminating one of the most important confounding factors in axon radius mapping (*Nilsson et al., 2012*). Second, gradient systems capable of producing relatively strong gradient pulses have been introduced in human scanners (*Jones et al., 2018*). Third, it has been shown that dMRI can be made specific to a particular water population restricted into long, yet micrometer-thin cylindrical objects by imparting high diffusion-weighting regimes (*McKinnon et al., 2017*; *Veraart et al., 2019*). Often, an axon is too narrow to yield a measurable diffusion-weighted MR signal decay, hence the popular use of 'sticks' (*Behrens et al., 2003*; *Kroenke et al., 2004*) when referring to axons (and possibly glial cell processes) within the context of biophysical modeling of white matter using dMRI.

The intuition behind promoting specificity to intra-axonal water comes from Callaghan's model (*Callaghan et al., 1979*) of diffusion inside infinitely narrow one-dimensional randomly-oriented cylinders, as applied to intra-neurite diffusion by *Kroenke et al. (2004)*. The spatial Fourier transform $e^{-D_a^\parallel (\mathbf{q}\hat{\mathbf{n}})^2 t}$ of the diffusion propagator (with respect to the diffusion wave vector $\mathbf{q}$) for a single stick as measured with MRI (*Callaghan, 1991*), averaged over the orientations $\hat{\mathbf{n}}$ of the sticks, yields the asymptotic *scale-invariant* power law $\overline{S} = \int \mathrm{d}\cos\theta \, e^{-D_a^\parallel q^2 t \cos^2\theta} \sim 1/b^\alpha$ as a function of the diffusion weighting parameter $b = q^2 t$ (*Le Bihan et al., 1986*), with the scaling exponent $\alpha = 1/2$. Evidently, this power law scaling should be only approximate, for $q \ll 1/r$, where $r$ is the cylinder radius. Its observation (*McKinnon et al., 2017*; *Veraart et al., 2019*) in the range $6\,\mathrm{ms}/\mu\mathrm{m}^2 \leq b \leq 10\,\mathrm{ms}/\mu\mathrm{m}^2$ is a manifestation of the *insensitivity* of dMRI to the transverse axonal dimensions on clinical scanners. However, for sufficiently strong diffusion weighting, the power law scaling eventually breaks down, and the dMRI measurement becomes sensitive to the axonal diameter.

Technically, this work addresses the detection and the interpretation of the *deviation* from the radius-insensitive $\alpha = 1/2$ power law signal behavior at the largest possible $b$ (by varying $q$ at fixed diffusion time $t$), in rat and human white matter. Indeed, either sensitivity of MR to a finite axonal radius, or a notable exchange rate between intra- and extra-axonal water at the clinical dMRI time scales $t \sim 100$ ms, would alter the very particular power law scaling (*Kroenke et al., 2004*; *Jensen et al., 2016*; *McKinnon et al., 2017*; *Veraart et al., 2019*).

Following theoretical considerations, we demonstrate the breaking of the power law scaling at very high $b$-values in ex vivo rodent brains, from which metrics associated with the axon radius distribution can be mapped quantitatively. Confocal microscopy of the rat corpus callosum (CC) validated

that (i) the signal arises mainly from the intra-axonal space, and (ii) the MR-derived axon radius metrics are in good quantitative agreement with those derived from histology. We then observe the same signal signatures in living human brain on the Connectom 3T scanner, that is, a high performance research scanner with a maximal gradient amplitude of 300 mT/m – a fourfold increase compared to state-of-the art clinical scanners (*Glasser et al., 2016*). Our findings both validate the mechanism with which axon radii are weighted in dMRI (*Burcaw et al., 2015*), and demonstrate the accuracy of which properties of the radius distributions can be estimated. After validating and evaluating our methodology in rat and human brain, we further discuss the impact of axon radius measurements in health and disease.

## Theory

### Power law scaling

In most biophysical models for diffusion in white matter, axons (and possibly glial cell processes) are represented by zero-radius impermeable 'sticks', characterized by locally one-dimensional diffusion, that is radial intra-axonal diffusivity $D_a^\perp \equiv 0$ (*Kroenke et al., 2004*; *Behrens et al., 2003*; *Jespersen et al., 2007*; *Jespersen et al., 2010*; *Fieremans et al., 2011*; *Sotiropoulos et al., 2012*; *Zhang et al., 2012*; *Novikov et al., 2014*; *Novikov et al., 2018*; *Novikov et al., 2019*; *Jensen et al., 2016*; *Reisert et al., 2017*; *McKinnon et al., 2017*; *Veraart et al., 2019*). The stick model then yields an asymptotic intra-axonal orientationally averaged signal decay,

$$\overline{S}(b) \simeq \beta\, b^{-\alpha} + \gamma, \quad bD_a^{\|} \gg 1, \tag{1}$$

with an intercept $\gamma$ (discussed below), the power law exponent $\alpha = 1/2$, and the coefficient $\beta = \sqrt{\pi/4} \cdot f/(D_a^{\|})^{1/2}$ where $f$ is the $T_2$-weighted axonal water fraction (*Veraart et al., 2018*; *Lampinen et al., 2019*) and $D_a^{\|}$ the parallel intra-axonal diffusivity. This particular signal decay only holds in the absence of extra-axonal signal, which is assumed to decay exponentially fast and, as such, to be fully suppressed at sufficiently high $b$-values (*McKinnon et al., 2017*; *Veraart et al., 2019*). Therefore, we restrict our in vivo analysis to $b > 6\,\mathrm{ms}/\mu\mathrm{m}^2$ (*Veraart et al., 2019*). Our lower bound on the $b$-value is significantly higher than previous predictions from Monte Carlo simulations (*Raffelt et al., 2012*), thereby minimizing the likelihood of residual extra-axonal signal contributions. For the ex vivo analysis, we increase this lower bound to $b = 20\,\mathrm{ms}/\mu\mathrm{m}^2$ to compensate for the reduced diffusivities in fixed tissue (*Shepherd et al., 2009*).

### Breaking of the power law

The following computations always assume that the signal is normalized to $S|_{b=0} \equiv 1$. Sensitivity of MR to either finite axon radius or notable exchange rate between intra- and extra-cellular water would break the $b^{-1/2}$-scaling at large $b$ as follows:

- Finite axon radius: A finite $D_a^\perp > 0$ results in a truncated power law:

$$\overline{S}(b) \simeq \beta\, e^{-bD_a^\perp + \mathcal{O}(b^2)}\, b^{-1/2} + f_{\mathrm{im}}, \tag{2}$$

with $f_{\mathrm{im}} \equiv S|_{b\to\infty} \geq 0$ the signal fraction of a fully restricted *immobile* water compartment (the so-called 'dot' compartment) (*Stanisz et al., 1997*; *Dhital et al., 2018*; *Tax et al., 2019*; *Veraart et al., 2019*). If $D_a^\perp = 0$, the power law offset (found from extrapolating the signal to $b \to \infty$), should give the fraction of the dot compartment, $\gamma \equiv f_{\mathrm{im}}$ (*Veraart et al., 2019*). However, for nonzero $D_a^\perp$ and finite $b$, the Taylor expansion of,

$$\overline{S} \simeq \beta\xi \cdot e^{-D_a^\perp/\xi^2} + f_{\mathrm{im}}, \quad \xi = 1/\sqrt{b} \tag{3}$$

around any finite point $\xi_0$ predicts the $\xi \to 0$ intercept $\gamma < f_{\mathrm{im}}$, *Figure 1*. The *always negative* difference $\epsilon = \gamma - f_{\mathrm{im}} < 0$ depends on $\beta$, $D_a$, and $\xi_0$; its maximal magnitude $|\epsilon_{\mathrm{max}}| = \beta\sqrt{2D_a^\perp/e} = f \cdot \sqrt{\frac{\pi}{2e} \cdot D_a^\perp/D_a^{\|}}$ is achieved at the curve's inflection point $\xi_*^2 = 2D_a^\perp$. Hence,

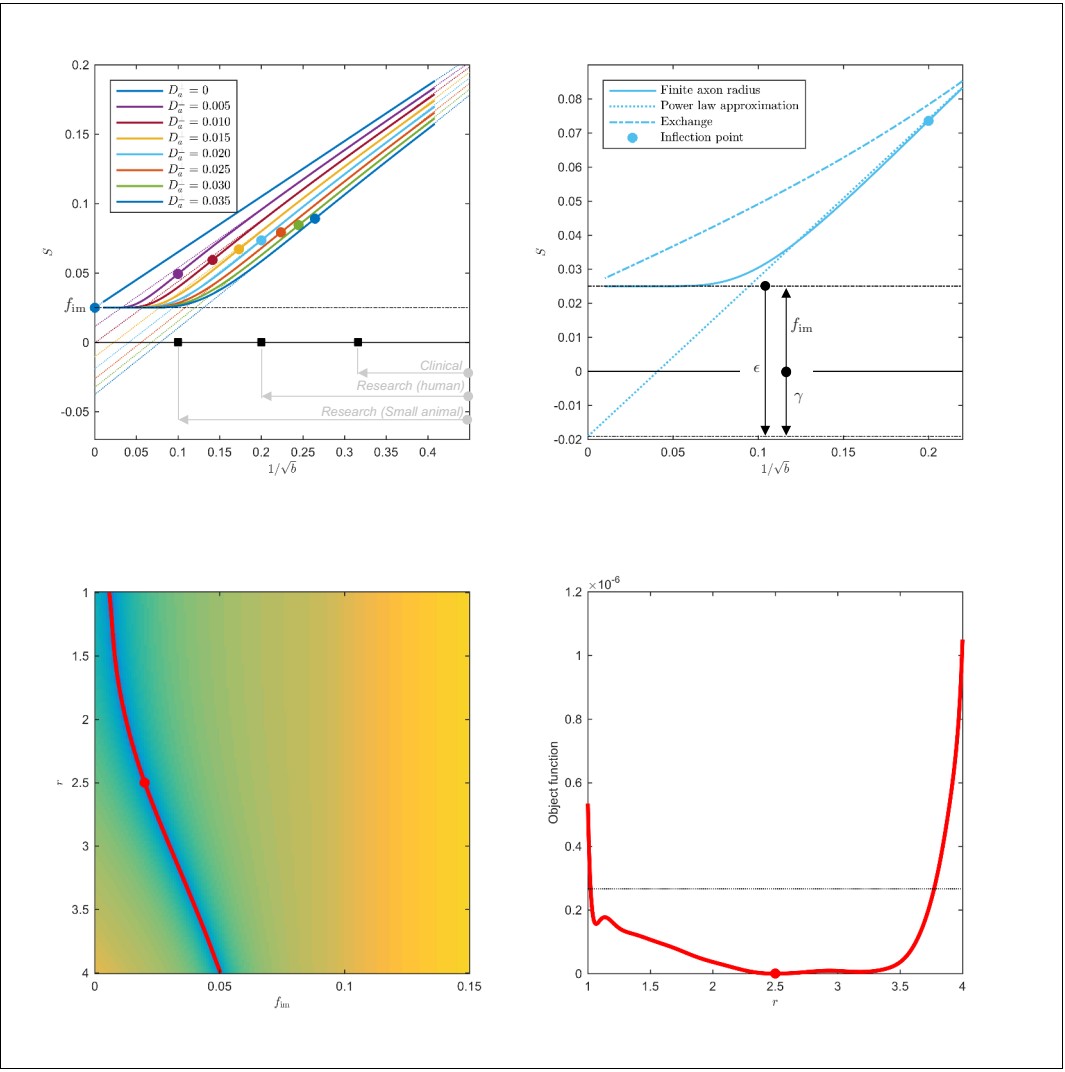

**Figure 1.** Breakdown of power law scaling: Top left: A nonzero $D_a^\perp$ would result in a truncated power-law signal decay. Although the resulting signal nonlinearity might be too subtle to be discerned within the achievable $b$-range, even for (pre-)clinical systems with strong diffusion-weighting gradients, the *concavity* of the curves plotted as function of $\xi = 1/\sqrt{b}$ for $\xi > \xi_* = \sqrt{2D_a^\perp}$ means that even the smallest $D_a^\perp$ will result in an extrapolated $\xi \to 0$ intercept $\gamma < f_{\mathrm{im}}$ when the power law, *Equation 1*, is used to approximately describe the signal in the delineated $b$-ranges. The intercept is maximally negative at the inflection point $\xi_*$ (colored dots), beyond which each curve becomes convex, and the negative intercept $\gamma$ of the linear extrapolation starts to decrease. In all plots here, diffusivities and $b$-values are expressed in $\mu m^2/ms$ and $ms/\mu m^2$, respectively. Top right: One representative curve ($D_a^\perp = 0.020$) is shown to highlight the differences between the physically plausible dot compartment $f_{\mathrm{im}} > 0$, and the intercept $\gamma$. The dot compartment is a positive signal fraction of a biophysical compartment, whereas the intercept is a parameter of the power-law approximation, *Equation 1*. Their difference $\epsilon$ depends on various parameters, including the axonal signal fraction, diffusivities, the axon radius, and the scan protocol. The predicted signal decay for the exchange model (dash-dotted; *Equation 4*) is convex in the entire $b$-range, where the signal decay for the finite axon radius model (dotted; *Equation 2*) is concave until the inflection point. Bottom: The optimization landscape of *Equation 3* shows a shallow valley, relative to the noise floor, for a simulation that mimics the human component of the study. (Bottom left) The valley is shown in a 2D projection of the landscape (shown as a function of radius instead of $D_a^\perp$, see *Equation 10*). (Bottom right) The fit objective function along the valley is shown (red line) in comparison to the noise floor (dashed line) with an unrealistically high SNR of 250 for the non-DW signal. The red dot indicates the ground truth value.

the lower bound $f_{\mathrm{im}} - |\epsilon_{\max}|$ for the $\xi \to 0$ intercept $\gamma$ may be negative. A negative $\gamma$ is biophysically implausible if the stick model holds, $D_a^\perp \equiv 0$; however, $\gamma < 0$ becomes a natural consequence of a finite $D_a^\perp > 0$ (and hence, of a finite axonal diameter), *Figure 1*, in the case when the extrapolated negative intercept overcomes the positive immobile fraction $f_{\mathrm{im}}$. Recently, $f_{\mathrm{im}}$ was shown to be negligible in healthy human white matter (*Dhital et al., 2018*; *Tax et al., 2019*; *Veraart et al., 2019*). Therefore, a negative intercept is a novel hallmark of MR sensitivity to the inner axon diameter, even if the signal scaling might appear linear as a function of $b^{-1/2}$ for $b$-ranges accessible on human MR scanners. Importantly, the finite axon radius model, *Equation 3*, is poorly conditioned as a result of which the simultaneous estimation of $f_{\mathrm{im}}$ and $D_a^\perp$ is practically impossible, especially for human MR experiments, see *Figure 1*. An accurate and precise measurement of $D_a^\perp$ depends on the prior knowledge of $f_{\mathrm{im}}$ and requires a dedicated measurement (*Dhital et al., 2018*; *Tax et al., 2019*).

- Exchange: The spherical integration of the two-compartment 'Kärger' model (*Kärger, 1985*) with a finite exchange rate $\mathcal{R} > 0$ yields approximately the following signal decay:

$$\overline{S}(b) \approx \beta\left( b^{-1/2} + c \cdot b^{-3/2} \right) + f_{\mathrm{im}}, \quad c \propto \mathcal{R} T_{\mathrm{E}}/D_e^\perp > 0, \tag{4}$$

with $D_e^\perp$ the radial diffusivity in the interstitial space, and $T_{\mathrm{E}}$, the echo time, during which exchange can happen. Importantly, *Equation 4* is *convex* as a function of $\xi = b^{-1/2}$.

The relative fit quality of the models (i.e., *Equations 1, 2, and 4*) to the dMRI signal decays can be evaluated qualitatively (convex versus concave shape) or statistically by means of the corrected Akaike information criterion (AICc) (*Burnham and Anderson, 2002*).

## From diffusivity to effective MR radius

The radial signal attenuation $S_c^\perp(r)$ inside the cylinder of radius $r$ in the Gaussian phase approximation (*van Gelderen et al., 1994*):

$$\ln S_c^\perp(r) \;=\; -\frac{2g^2 r^4}{D_0} \sum_{m=1}^{\infty} \frac{t_c}{\alpha_m^6(\alpha_m^2 - 1)} \cdot \left[ 2\alpha_m^2 \frac{\delta}{t_c} - 2 + 2e^{-2\alpha_m^2 \delta/t_c} + 2e^{-2\alpha_m^2 \Delta/t_c} - e^{-2\alpha_m^2(\Delta - \delta)/t_c} - e^{-2\alpha_m^2(\Delta + \delta)/t_c} \right] + \mathcal{O}(g^4)$$
$$\equiv -b D_a^\perp(r) + \mathcal{O}(b^2), \tag{5}$$

with $b = g^2 \delta^2 (\Delta - \delta/3)$ and $t_c = r^2/D_0$ defines the connection between the intra-axonal radial diffusivity $D_a^\perp$ and the radius $r$. Here, $D_0$ is the diffusivity of the axoplasm, $g$ the gradient of the Larmor frequency, $\alpha_m$ is the $m^{\mathrm{th}}$ root of $\mathrm{d}J_1(\alpha)/\mathrm{d}\alpha = 0$, where $J_1(\alpha)$ is the Bessel function of the first kind, and $\delta$ and $\Delta$ are the gradient duration and separation, respectively (*Stejskal, 1965*).

In the long-pulse limit, that is when $\delta \gg t_c$, the dependence on $\Delta$ drops out (*Neuman, 1974*), and *Equation 5* approaches the Neuman's limit

$$\ln S_c^\perp(r) = -\kappa\, r^4, \quad \kappa = \frac{7}{48} \frac{g^2 \delta}{D_0}, \quad \delta \gg t_c. \tag{6}$$

This limit practically applies to the majority of axons. Importantly, the attenuation is proportional to the fourth power of the radius $r$ and, as such, it is very weak for narrow axons. Hence the low sensitivity of dMRI to the inner axon diameter.

For an unknown distribution $h(r)$ of axons with radii $r$, the total intra-axonal signal attenuation becomes a volume-average over the histogram bins $r_i$ (*Packer and Rees, 1972*):

$$S^\perp[h(r)] \simeq \frac{\sum_i h(r_i) r_i^2 S_c^\perp(r_i)}{\sum_i h(r_i) r_i^2} = \frac{\langle r^2 S_c^\perp(r) \rangle}{\langle r^2 \rangle}, \tag{7}$$

such that the signal contribution of an axon scales quadratically with its radius $r$. The Taylor expansion of the net signal attenuation $S^\perp$ demonstrates the sensitivity of the dMRI signal to the distribution's higher order moments:

$$S^\perp[h(r)] = \langle r^2(1 - \kappa r^4 + \mathcal{O}(r^8))\rangle/\langle r^2\rangle \approx 1 - \kappa\langle r^6\rangle/\langle r^2\rangle$$
$$\approx e^{-\kappa r_{\mathrm{eff}}^4} \equiv S_c^\perp(r_{\mathrm{eff}}),$$

(8)

where the effective axon radius:

$$r_{eff} \equiv (\langle r^6\rangle/\langle r^2\rangle)^{1/4}$$

(9)

captures the contribution from the whole distribution $h(r)$ in a single metric (*Burcaw et al., 2015*). The ability to represent the whole distribution by the ratio of its 6$^{\text{th}}$ and 2$^{\text{nd}}$ moments relies on almost all axons falling into the Neuman's limit, *Equation 6*. Representing $S_c^\perp(r_{\mathrm{eff}}) \equiv e^{-bD_a^\perp}$, we can calculate

$$r_{MR} = \left(\frac{48}{7}\delta(\Delta - \delta/3)D_0 D_a^\perp\right)^{1/4},$$

(10)

the MRI estimate of $r_{eff}$ after estimating $D_a^\perp$ from the orientation-averaged signal using *Equation 2*.

Note that the effective radius, *Equation 9*, is heavily weighted by the tail of $h(r)$. Physically, this happens due to the combination of the weak NMR signal attenuation by small radii, $\ln S \sim -r^4$, in the diffusion-narrowing (Neuman's) regime (*Neuman, 1974*), and of the subsequent volume-weighting that emphasizes the thickest axons by an extra factor of $r^2$ (*Packer and Rees, 1972*; *Alexander et al., 2010*). The error associated with these modeling assumptions is discussed in the *Results* section.

## Results

### Simulations

#### Accuracy

First, we evaluate the accuracy of axon radius mapping as a function of $r$ for axon radius distributions extracted from histology; *Figure 2* (left and middle panels). We used a simulation framework based on the matrix formalism for diffusion signal attenuation within fully restricted cylinders (*Callaghan, 1997*), as implemented in the MISST toolbox (*Drobnjak et al., 2010*), while mimicking the entire experimental setup, for both the human and preclinical experiments.

In the case of diffusion restricted in a single cylinder with radius $r$, the error in the estimated radius $\hat{r}$ increases with $r$. Indeed, the missing higher-order $\mathcal{O}(g^4)$ corrections to *Equations (5)-(6)* set an upper bound on the achievable accuracy for large axons, as estimated recently (*Lee et al., 2018*).

The combined error in the estimation of $r_{\mathrm{eff}}$ associated with the approximations made in *Equation 6* and *Equation 8* is only 5% for the human set-up when considering the axon radius distribution provided by *Aboitiz et al. (1992)*; the distributions of *Caminiti et al. (2009)* result in a subpercent error. Additionally, we show the errors in the estimation of $r_{\mathrm{eff}}$ for the axon caliber distributions that were observed in our different histological sections while considering the scan parameters from our fixed tissue experiments. The shorter diffusion timings increase the approximation errors, leading to an underestimation up to 9%.

#### Feasibility and precision

*Figure 2* (right panel) shows a theoretical lower bound on the 95% confidence interval in the voxelwise estimation of $D_a^\perp$ from *Equation 2*, as predicted using a Cramér-Rao lower bound analysis (*Kay, 1993*). Using the dependence $D_a^\perp \approx D_a^\perp(r_{\mathrm{eff}})$, *Equation 8*, that approximately identifies $r_{\mathrm{eff}}$ with the single cylinder radius in van Gelderen's model, can be used to translate the lower bound on $D_a^\perp$ to that on $r_{\mathrm{eff}}$.

Notably, it follows from *Figure 2*, that an estimate of $D_a^\perp$ exceeds zero with a statistical threshold of $p > 0.05$, if the corresponding $r_{\mathrm{eff}} > 1.41\,\mu\mathrm{m}$ and $r_{\mathrm{eff}} > 0.76\,\mu\mathrm{m}$, when mimicking the diffusion acquisition and SNR on the Siemens Connectom ($G_{\mathrm{max}} = 300\,\mathrm{mT/m}$) and Bruker Aeon ($G_{\mathrm{max}} = 1500\,\mathrm{mT/m}$) MR scanners, respectively. In comparison, for a typical acquisition on a modern clinical scanner with $G_{\mathrm{max}} = 80\,\mathrm{mT/m}$, this lower bound is 3.2 $\mu$m (*Veraart et al., 2019*).

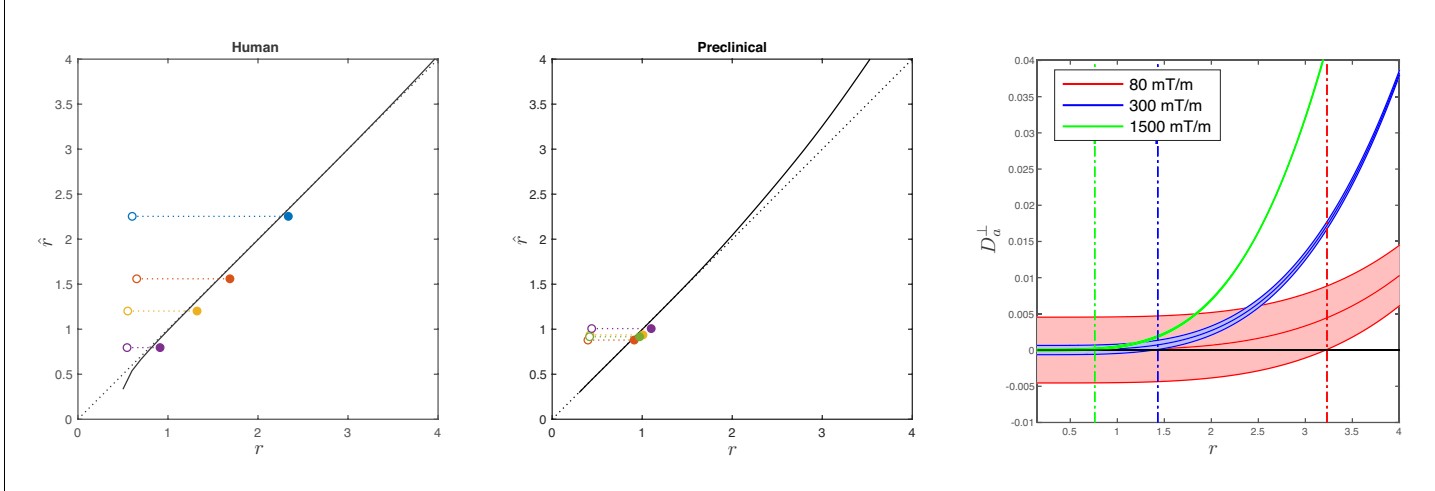

**Figure 2.** Simulations on accuracy and precision of MR-based axon radius mapping. First, the left and middle panel show the difference between the estimated, $\hat{r}$, and theoretical, $r$, effective MR radius associated with various realistic axon caliber distributions (*solid dots* with different color for different distributions) for the clinical and preclinical setups, respectively. Axon caliber distributions were adopted from *Aboitiz et al. (1992)* and *Innocenti et al. (2015)* for the clinical simulations (see *Figure 7*), whereas various axon distributions (see *Figure 4*) derived from our own histology were used for the preclinical simulation. The average radii, $\bar{r}$, of the axon caliber distribution are shown for comparison (*open dots*). Additionally, the accuracy of the framework for a system with single cylinder with radius $r$ is shown (*black line*). Second (right figure), the feasibility to measure $D_a^\perp$ with statistical significance in case of scan settings and SNR for the Connectom (300 mT/m; blue), Aeon (1500 mT/m; green) protocol, respectively. For comparison, we also assessed the feasibility for the Prisma protocol as described in *Veraart et al. (2019)* (80mT/m; red). The shaded areas illustrate the 95% confidence intervals derived from Cramér-Rao lower bound analysis of model, *Equation 2* with $f_{\rm im} = 0$. The corresponding minimal cylinder radius $r$ that allows for the observation of significant $D_a^\perp(r)$, $r = 0.76\,\mu{\rm m}$, 1.41 μm and 3.24 μm for Aeon, Connectom, and Prisma, respectively, is indicated by the vertical lines. In all plots, diffusivities and radii are expressed in $\mu{\rm m}^2/{\rm ms}$ and $\mu{\rm m}$, respectively.

## Preclinical data

### Dot compartment

Because $f_{\rm im}$ has been reported to be significant in fixed tissue by *Stanisz et al. (1997)*, we first have to estimate the signal fraction of the immobile water compartment $f_{\rm im}$ in the three fixed brain samples from a dedicated MRI acquisition (see Materials and methods). The estimate $\hat{f}_{\rm im}$ is in the range of 8 to 17% with a median value of 13%. The range is defined here by the 5[th] and 95[th] percentile of the distribution of the estimated dot fractions in all CC voxels, across the three samples.

### Breaking the power law

*Figure 3* shows the signal decay, averaged across all CC voxels, based on diffusion measurements in the three rat brain samples with $b$ up to $100\,{\rm ms}/\mu{\rm m}^2$. We notice that an extrapolation to infinite $b$, that is $1/\sqrt{b} \to 0$, yields a small but significant *negative* offset $\gamma$, of the order of a few per cent of the non-attenuated $S|_{b=0}$ signal, in all three samples after subtracting the $\hat{f}_{\rm im}$ from the diffusion measurements.

We re-evaluate the validity of a perfect stick assumption in the high $b$-regimes using a AIC analysis. To study fit robustness with respect to the number of degrees of freedom by considering the full, nested, and extended models to *Equation 1*. Specifically, we evaluated the following models:

i. $f_{\rm im} + \beta b^{-\alpha}$
ii. $f_{\rm im} + \beta b^{-1/2}$;
iii. $f_{\rm im} + \beta e^{-bD_a^\perp} b^{-1/2}$
iv. $f_{\rm im} + \beta\left(b^{-1/2} + c \cdot b^{-3/2}\right)$
v. $\beta b^{-\alpha}$
vi. $\beta b^{-1/2}$
vii. $\beta e^{-bD_a^\perp} b^{-1/2}$
viii. $\beta\left(b^{-1/2} + c \cdot b^{-3/2}\right)$

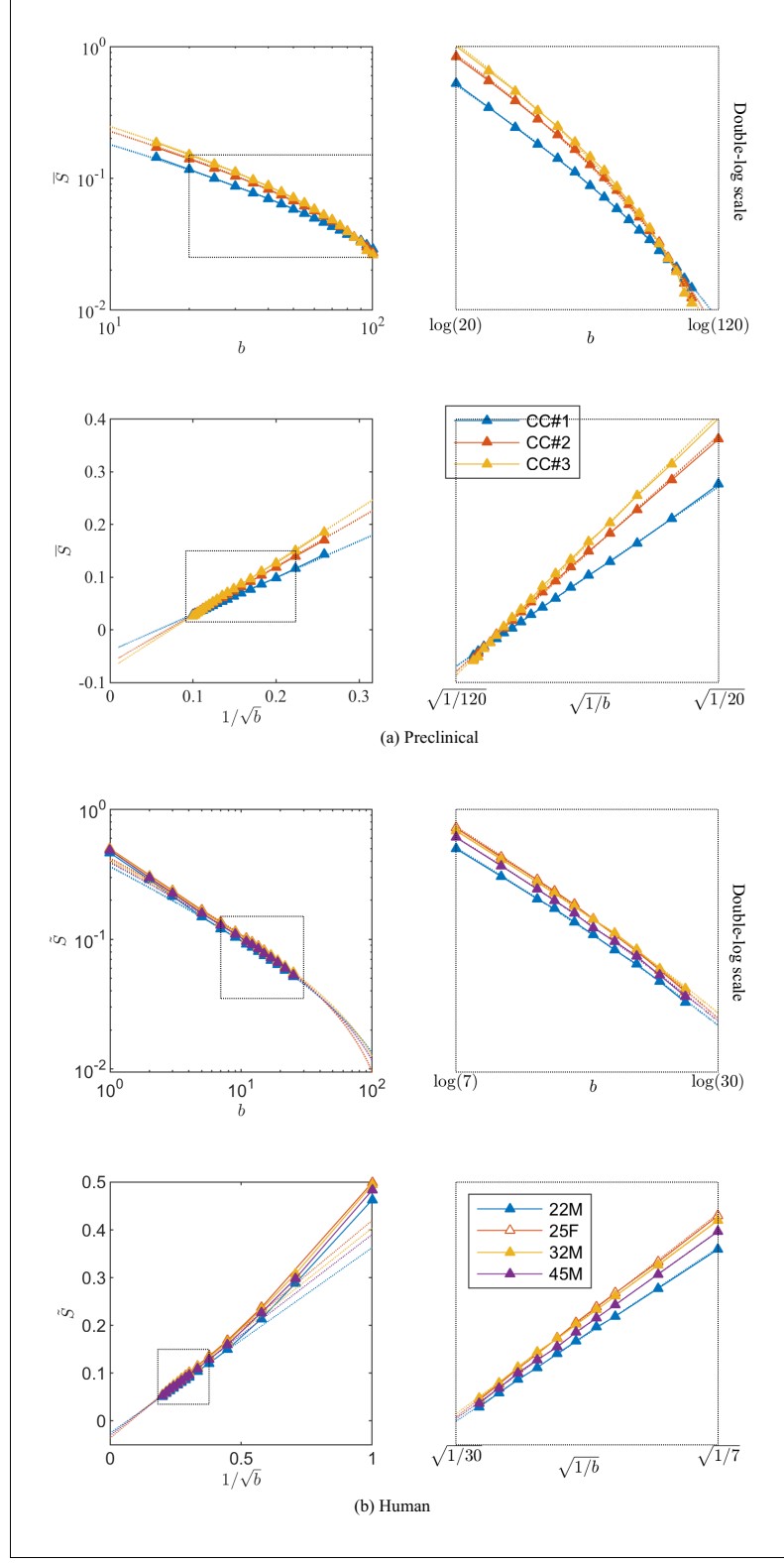

**Figure 3.** Breaking of the power law. The ROI- and spherically averaged signal decay is shows for the different fixed samples of the rat CC (a) and human subjects (b) and as a function of $1/\sqrt{b}$ and on a double logarithmic scale. The data deviate from the power law scaling with exponent 1/2 that is predicted by the stick model (i.e. nonlinear signal decay in log-log plot), thereby demonstrating sensitivity of the signal to the radial intra-axonal signal. The fits of *Equation 1* are shown in dashed lines. In all plots, $b$ is expressed in $\mathrm{ms}/\mu\mathrm{m}^2$.

Our analysis shows that a truncated power law (vii), which explicitly accounts for $D_a^\perp > 0$ (and hence does not require a negative intercept parameter), yet sets $f_{im} = 0$, fits the experimental data significantly better than pure power law forms (models (i), (ii), (v), and (vi)) (the difference in $\mathrm{AICc} < 2$), with or without an offset $\gamma$, if the immobile (dot) compartment is corrected for, that is when using $\overline{S}_\star(b) = \overline{S}(b) - \hat{f}_{im}$. Without *dot*-correcting the ex vivo data, the power law form (ii) with an intercept outperforms the other models. In that case, the intercept $\gamma$ is negative, while $f_{im}$ is defined to be positive. Hence, the intercept encodes both the still water fraction and a negative offset to the intercept associated with the sensitivity to the axon diameter, such that the overall $\gamma < 0$.

## Axon radius estimation and histological validation

Axon radii were estimated from the diffusion MRI data for the different CC ROIs (*Figure 4*) along with the axon radius distributions extracted histologically, *Figure 4*. The errors between the associated tail-weighted effective radii and MR-derived $r_{MR}$ vary between 5 and 21% in the different ROIs. Notably, a consistent residual overestimation was observed, whereas the previous simulations (*Figure 2*) predicted an underestimation.

To further examine the correlations between the MR-derived parameters and underlying microstructure, we analyzed 16 patches with the same size as an imaging voxel, that is $100 \times 100 \ \mu m^2$ within the genu of the CC of the second sample, CC#2. The confocal light microscopy images of two of those patches are shown for the various stainings, *Figure 5*. Two notable biological components other than axons were highlighted, namely, astrocytic processes and (neuronal or glial) cell bodies, which were found to have volume fractions of about 5%. Although the radius of the astrocytic processes cannot be measured due to their random orientations w.r.t. the image plane, it is clear that some of the processes have a large diameter, for example up to 7 $\mu m$ in the first patch. The average cell body radius is 2.6 $\mu m$, with an effective radius of 4.3 $\mu m$. It is worth highlighting that the $T_2$-weighted signal fraction of both cell types remains unknown since the corresponding

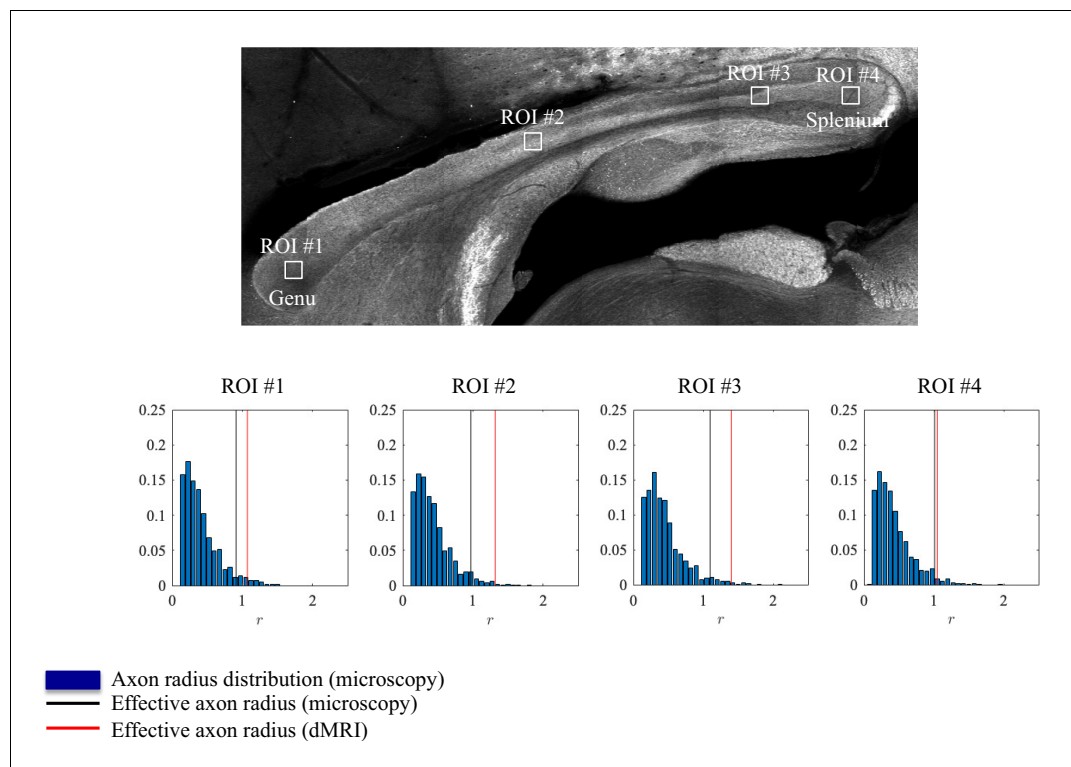

**Figure 4.** Histological validation, part I. The axon radius distributions for different ROIs of rat CC#1 are shown (blue bars).The associated tail-weighted effective radii are shown in the black lines, whereas the corresponding MR estimates are shown by the red lines. In all plots, $r$ is expressed in $\mu m$.

relaxation times are unknown. This unknown difference between compartmental volume (histology) and signal (MRI) fractions remains the Achilles's heel of comparisons between MRI measurements and histological evaluations of tissue microstructure.

Within each of the 16 patches, we extracted the axon radii distribution and derived the average $\bar{r}$ and effective radius $r_{\mathrm{eff}}$. The box plots of those metrics are shown in *Figure 5*. The median $\bar{r}$ and median effective radius $r_{\mathrm{eff}}$ across all patches are 0.61 and 1.06 $\mu\mathrm{m}$, respectively. In comparison, the median $r_{\mathrm{MR}}$, derived from dMRI in 16 voxels within the genu of the CC, is 1.16, 1.10, and 1.19 $\mu\mathrm{m}$ for the three rat samples, respectively. The median MR-derived effective axon radius is between 81 and 97% larger than the median $\bar{r}$, whereas the error to the median $r_{\mathrm{eff}}$, as derived from histology, is only 3.4 to 12.8%.

## Parameter maps

ROI measurements provided robust estimation, but a remaining question is whether dMRI could be used to map the effective MR radius in a voxelwise manner. *Figure 6* shows the maps of the MR-derived effective axon radii for all three rat CC's. The maps appear smooth with very few outlier voxels, suggesting that the estimation is robust even when voxelwise data is used. Furthermore, the qualitative trends are in good agreement with previously reported observations of larger axons in the body of the CC in comparison to the genu and splenium (*Barazany et al., 2009*). Inter-subject variability is not very large and can be attributed to slightly different slice positions. When computing the effective radius of the CC-averaged signal $\bar{r}_{\mathrm{MR}}$, the intersubject variability nearly nullifies. Indeed, we estimate $\bar{r}_{\mathrm{MR}} = 1.22, 1.25$ and 1.25 μm in the three samples, respectively.

## Towards human applicability
### Breaking the power law

To assess the applicability of this approach in more realistic settings available for human imaging, experiments were performed in human subjects on the Connectom scanner, which is capable of producing 300 mT/m gradient amplitudes. The dMRI signal decay curves, averaged across all WM

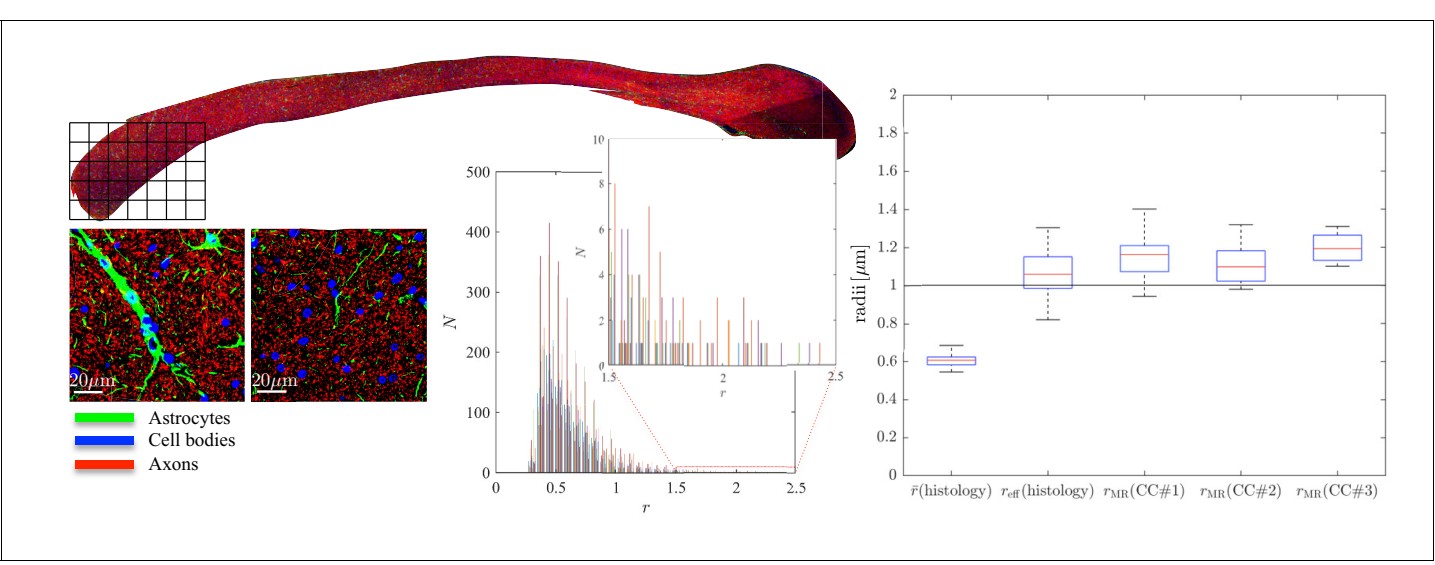

**Figure 5.** Histological validation, part II. (left) For a second fixed brain sample, CC#2, the confocal microscopy images, stained for neurofilaments (red), astrocytes (green), and cell bodies (blue), are shown for two representative $100 \times 100\,\mu\mathrm{m}^2$-patches that are positioned within the Genu (microscopy image of CC shown for ROI positioning). The abundance of astrocytes and cell bodies, both representing 5% of the volume, is clear in both patches. The astrocytic processes can have a large diameter, up to 7 μm in the first patch. A detailed analysis of the radius distribution of the astrocytic processes is not possible due to their random orientation w.r.t. the image plane. (middle) Axon radius distributions for all 16 patches of the Genu (each patch has different color in the bar plot). (right) Boxplots represent the distribution of the average and effective radius of the axon radii distribution that were extracted from each of the 16 patches within the genu. The effective radius $r_{\mathrm{eff}}$ is larger than the average $\bar{r}$, respective medians are 1.06 and 0.61 μm. The boxplots for the MR-derived axon radius measurements for 16 MR voxels within the genu for the three fixed CC samples are also shown. In all plots, radii are expressed in $\mu\mathrm{m}$.

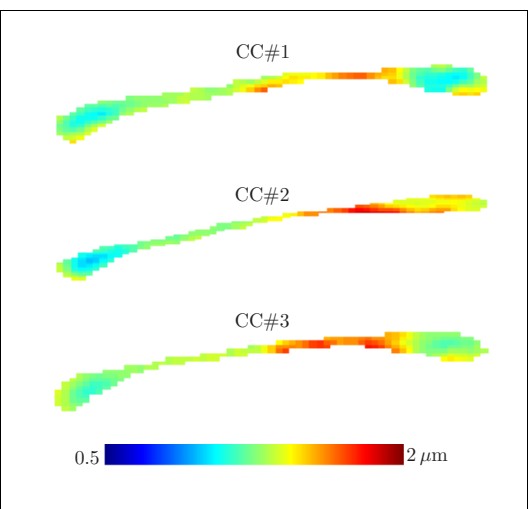

**Figure 6.** Effective radii in the CC. Maps of the effective radii derived from diffusion MR data, for the 3 samples of the rat CC.

voxels, with $b$-values up to 25 ms/μm$^2$ for the four human subjects are shown in *Figure 3*. Importantly, we find that — in excellent correspondence with the previous preclinical data — the linear extrapolation of the signal decay as a function of $1/\sqrt{b}$ to $1/\sqrt{b} \rightarrow 0$ produces a significant negative offset $\gamma$ in all subjects.

Note that the dot compartment was not measured directly, because previous dedicated studies revealed a negligible dot compartment, that is $f_{\mathrm{im}} = 0$, in the healthy white matter (*Dhital et al., 2018*; *Veraart et al., 2019*; *Tax et al., 2019*); see Discussion.

The AICc analysis of various models demonstrated that also for the human white matter, the truncated power law (vii) with $D_a^\perp > 0$ and negligible dot compartment $f_{\mathrm{im}} = 0$ fits the data significantly better than pure power law forms, with or without intercept. However, this statistical analysis cannot be interpreted as a data-driven justification for $f_{\mathrm{im}} = 0$ because of the degeneracy of *Equation 3*, as highlighted in *Figure 1*.

## Comparison with histology

Since direct histological evaluation in volunteers is unfeasible, we turn to validate the MR-derived metrics in humans with previously reported literature of human corpus callosum microstructure (*Aboitiz et al., 1992*; *Innocenti et al., 2015*). In *Figure 7*, the MR-derived metrics were directly compared with axon radius distributions of multiple histological studies (*Aboitiz et al., 1992*; *Innocenti et al., 2015*). Various reports and histological studies show a good correspondence for the bulk of the distributions, represented by the average radius $\bar{r}$, that is the average radius $\bar{r}$ only ranges between 0.54 and 0.69 μm. In histological samples, the corresponding effective radius $r_{eff}$ dominated by large axons, shows strong variability. Indeed, compared to $\bar{r}$, $r_{eff}$ varies more than three-fold, from 0.91 to 2.9 $\mu$m.

The four human subjects show good correspondence in terms of $r_{MR}$. In *Figure 7*, we show the individual and combined distributions describing $r_{MR}$ for all voxels in the midbody of the CC for all four subjects. It is apparent that the combined distribution falls almost entirely within the range spanned by $r_{eff}$-values as predicted from histology – even without introducing a putative axonal shrinkage factor (maximally 35% [*Aboitiz et al., 1992*], and typically within 10% [*Tang et al., 1997*]).

## Parameter distribution and maps

In *Figure 8*, the distribution and map of $D_a^\perp$ for WM voxels in all human subjects, estimated using the ODF-independent model, *Equation 2* with $f_{\mathrm{im}} = 0$, are shown. Considering the statistical bound from *Figure 2*, it is to be expected that the estimated $D_a^\perp$ is biophysically meaningful for the vast majority of WM voxels for the Connectom scanner (*Figure 8* shows a representative map of $D_a^\perp$ and associated $r_{\mathrm{MR}}$ for a single subject of the Connectom cohort), whereas similar measurements on a modern clinical scanner result in a biophysically implausible negative $D_a^\perp$ in approximately 35% of all WM voxels. Note that the data from a clinical scanner (Siemens Prisma with 80 mT/m gradients) are adopted from our recent work (*Veraart et al., 2019*). This suggests that estimating $D_a^\perp$ and the associated effective axonal radius $r_{\mathrm{MR}}$ is only possible on MR systems with ultra-strong gradients (*Jones et al., 2018*; *Huang et al., 2015*). The spatial variability as well as the observed asymmetry between the hemispheres in, for example, the occipital lobes was noted for all subjects. However, our cohort is too small and not sufficiently characterized to study the whole brain characterization or the role of lateralization in large axons of human brain (*Eichert et al., 2019*; *Liewald et al., 2014*; *Lebel and Beaulieu, 2009*).

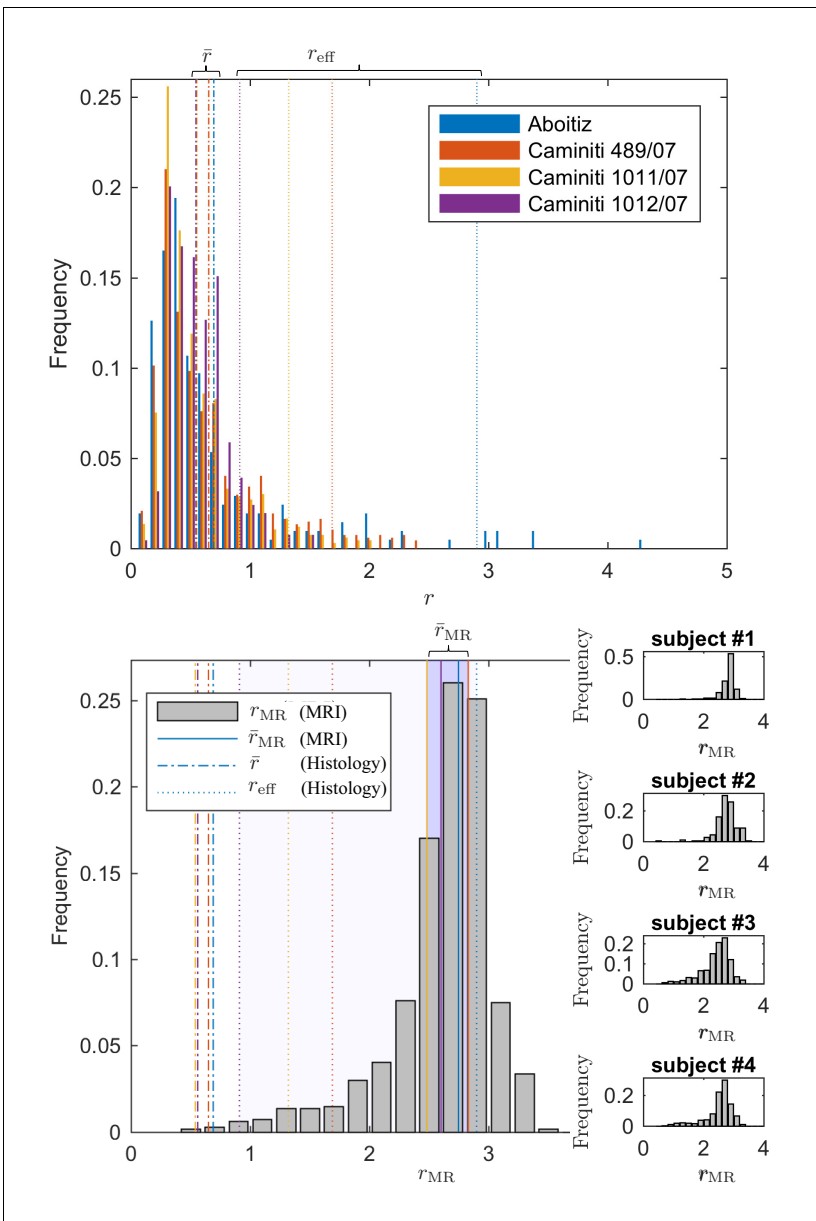

**Figure 7.** Comparing the effective radius from histology and in vivo dMRI. (top - histology) Axon radius distributions of multiple histological studies and human CC samples show a good correspondence for the bulk of the distribution, represented by the average radius $\bar{r}$ (dashed-dotted lines). Due to mesoscopic fluctuations of the large axons in histological samples, the corresponding effective radius $r_{eff}$ dominated by large axons, shows strong variability (dotted lines). (bottom - MRI) The four Connectom subjects show good correspondence in terms of $r_{eff}$. The distribution describing $r_{eff}$ for all voxels in the midbody of the CC for all four subjects falls almost entirely within the range spanned by values predicted by histology, with no need to account for potential shrinkage (*Horowitz et al., 2015*) during tissue preparation. In all plots, radii are expressed in $\mu$m.

## Gray matter

It is worth examining the power law scaling also in areas outside the white matter. Therefore, *Figure 9* shows the diffusion-weighted signal decay, averaged over all cortical gray matter (GM) voxels as a function of $b$ in the human subjects. The signal scaling in the WM is shown for qualitative comparison. The non-linear scaling of the isotropically-averaged signal as a function of $1/\sqrt{b}$ of all human subjects indicates strong deviations from the 'stick' model in the cortical GM, (*McKinnon et al., 2017*; *Palombo et al., 2019*). Accounting for a finite neurite radius, *Equation 2*, does not describe

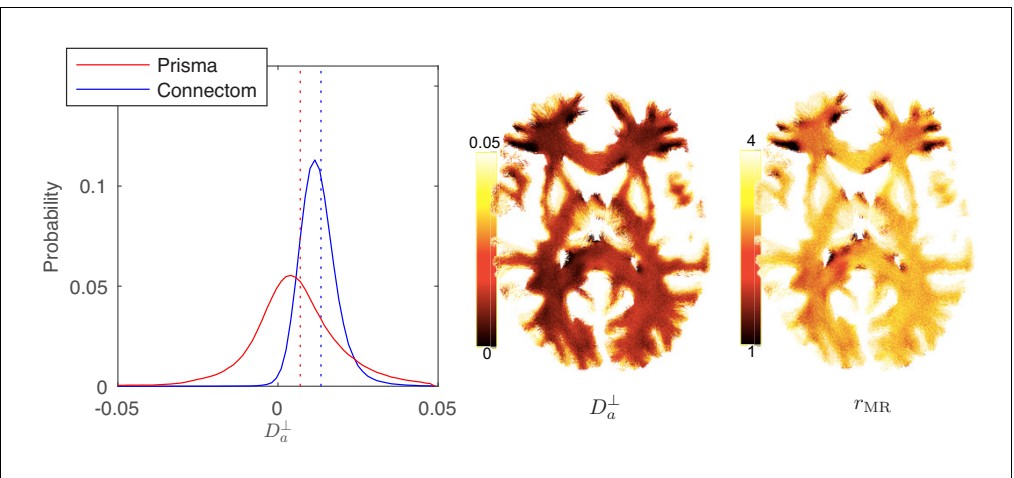

**Figure 8.** Distribution and maps of $D_a^{\perp}$ and $r_{\mathrm{MR}}$. (left) The distribution of $D_a^{\perp}$ estimated via **Equation 3** for all WM voxels (all scanner-specific subjects pooled) shown for both scan set-ups. In agreement with **Figure 2**, Prisma (80mT/m) data shows a much lower precision for the estimator of $D_a^{\perp}$. Despite the small yet positive mean value and the associated negative offset $\gamma$ in **Figure 3**, a large number of WM voxels yield biophysically implausible $D_a^{\perp}<0$ values. Precision drastically improves on the Connectom scanner (300 mT/m). (right) Maps of $D_a^{\perp}$, and of the effective MR radius heavily weighted by the tail of axonal distribution (**Figure 7**), for a single subject. Here, $r_{\mathrm{MR}}$ is derived from $D_a^{\perp}$ via **Equation 10**. In all plots, diffusivities and radii are expressed in $\mu\mathrm{m}^2/\mathrm{ms}$ and $\mu\mathrm{m}$, respectively.

the data well either. Instead, the *convex* signal decay as a function of $1/\sqrt{b}$ at high $b$-values is in good agreement with the anisotropic exchange model that we derived from the expansion of the anisotropic Kärger model in the powers of inverse $b$, **Equation 4**. Both the finite radius and exchange model predictions are shown in the absence of an immobile water fraction. The exchange model fits the data better than all other evaluated models in all subjects according to an AIC analysis (data not shown). The residence time within the neurites $1/\mathcal{R}$ varies from approximately 10 to 15 $\mathrm{ms}$ or 20 to 30 $\mathrm{ms}$ if we assume $D_e^{\perp} = 1\,\mu\mathrm{m}^2/\mathrm{ms}$ or $D_e^{\perp} = 0.5\,\mu\mathrm{m}^2/\mathrm{ms}$, respectively. Dedicated experiments are required for a more precise measurement of the exchange rate.

## Discussion

### What do we measure with dMRI?

Noninvasively estimating metrics associated with axon radius distributions is a formidable task, yet it could have a strong impact for numerous areas of research including neuroscience, biomedicine and even for clinical research and applications. Histological studies have extensively reported axon diameters $2r$ to be in the range $0.5 - 2$ µm for human WM (*Aboitiz et al., 1992*; *Caminiti et al., 2009*; *Liewald et al., 2014*; *Tang et al., 1997*), with only 1% of all axons having a diameter larger than 3 µm (*Caminiti et al., 2009*). A vigorous debate has emerged in the MRI and neuroanatomy communities as in vivo, MRI-derived axon diameters are reported to fall within the range $3.5 - 15$ µm (*Alexander et al., 2010*; *Horowitz et al., 2015*; *Huang et al., 2015*). On the MRI side, the discrepancy has been attributed to the long diffusion pulses that strongly reduce the signal attenuation of protons restricted in a narrow cylinder (*van Gelderen et al., 1994*; *Burcaw et al., 2015*). Therefore, the time-dependence of extra-axonal diffusion $D_e^{\perp}(t)$ (*Burcaw et al., 2015*; *Fieremans et al., 2016*; *Lee et al., 2018*), and the undulation or along-axon caliber variation (*Nilsson et al., 2012*; *Brabec, 2019*; *Özarslan et al., 2018*; *Lee et al., 2019*), potentially overshadow the relatively small $D_a^{\perp}$.

On the other hand, shrinkage during tissue fixation has been suggested as a potential shortcoming of histology (*Barazany et al., 2009*; *Horowitz et al., 2015*), implying that in vivo axons are thicker than their histologically reported values.

This study aimed at investigating what the dMRI signal can measure in terms of axon radius, as well as provide insight into the aforementioned debate. Our wide range of diffusion weightings in

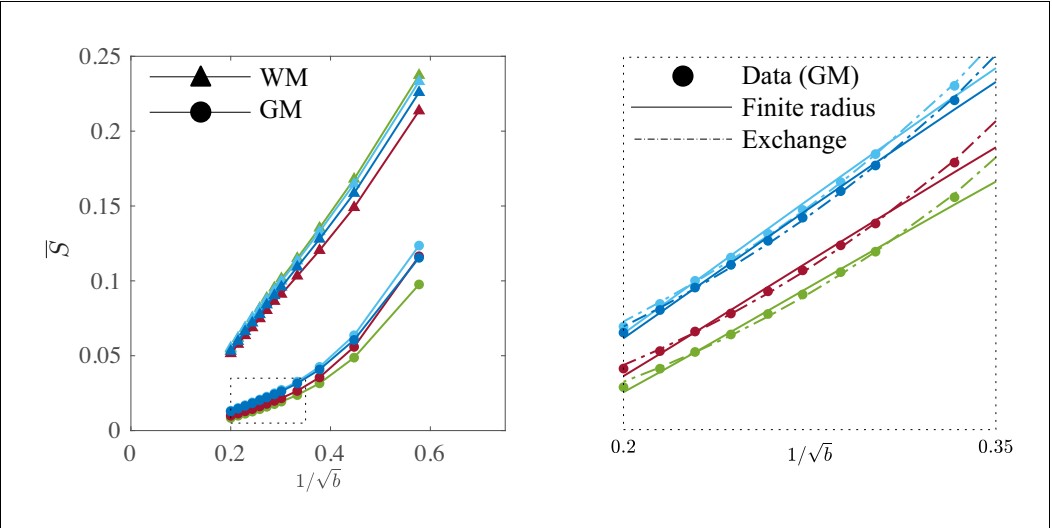

**Figure 9.** Signal decay in the GM. The spherically-averaged signal decay in the WM and GM is shown for all human subjects as a function of $1/\sqrt{b}$. The consistent non-linear scaling of the signal as a function of $1/\sqrt{b}$ demonstrates deviations from the 'stick' model in the cortical GM. In contrast to the WM, the convex signal decay in the GM is better described by an anisotropic exchange model of two compartments (*Equation 4*), than the finite radius model (*Equation 3*). In all plots, $b$ is expressed in $\text{ms}/\mu\text{m}^2$.

both human and preclinical dMRI enables a suppression of the extra-axonal contribution (that otherwise biases the radii *Burcaw et al., 2015*; *Fieremans et al., 2016*), thereby allowing us to shed light on this controversy. We claim that the effective MR radii measured in this study ($r_{\text{MR}}$) quantitatively agree with those derived from histology — to the extent that histology correctly captures the tail of the axonal radii distribution $h(r)$. That is, $r_{\text{MR}}$ obtained from dMRI appears to be a self-averaging quantity in each imaging voxel, as large MRI voxels ensure that the moments of $h(r)$ sampled from a voxel represent well the 'true' underlying $h(r)$ in that WM region, so that the spatial variations in $r_{\text{MR}}$ stem mainly from genuine biological variations of the tails of axon distributions across the brain.

## Mesoscopic fluctuations

Histology-derived $r_{\text{eff}}$ are prone to *mesoscopic fluctuations* due to small sampling sizes, *Figure 7*. Despite a good correspondence of the *bulk* of axon radii distributions obtained from different histological studies and samples (*Aboitiz et al., 1992*; *Caminiti et al., 2009*; *Liewald et al., 2014*; *Tang et al., 1997*), the *tail* of the distribution is typically coarsely sampled, with only a few spikes representing the occasional observation of large axons within the relatively small histological sections, *Figure 7*. It is precisely for the detectable large $r$, that the relative fluctuations for the bin counts $N_i$ are observed for bin values of $N_i \sim 1$ (Table 2 of *Aboitiz et al. (1992)* and *Figure 5*), according to the Poissonian statistics governing each $N_i$. Not surprisingly, $r_{\text{eff}}$ derived from discrete histological histograms exhibits strong fluctuations, as depicted by dotted vertical lines in *Figure 7a* and the error bars in *Figure 5*.

### Humans

Although the average radius $\bar{r}$, as reported in human literature (*Aboitiz et al., 1992*; *Innocenti et al., 2015*), only ranges between 0.54 and 0.69 $\mu$m, the corresponding $r_{\text{eff}}$ varies from 0.91 to 2.9 $\mu$m. In comparison, the average dMRI-derived $r_{MR}$ estimated from the four Connectom data sets within the same region-of-interest, the midbody of the CC, only varied from 2.48 to 2.82 $\mu$m (*Figure 7b*).

### Rodents

The average radius, as measured in this study, varies between 0.54 and 0.68 $\mu$m across the 16 patches of the genu of the CC, while the associated $r_{eff}$ varies from 0.81 to 1.30 $\mu$m. The MR-derived

effective axon radii $r_{MR}$ vary similarly, that is 0.94 to 1.4 $\mu$m across several voxels within the genu of the CC for all three scanned samples.

For dMRI, the variability in the estimation of $r_{MR}$ is determined by thermal MRI noise, and genuine anatomical – inter-voxel and inter-subject – differences. For human MRI, the mesoscopic fluctuations are much weaker, due to the large MRI voxels in comparison to the histological samples. Indeed, the variance in the estimation of $r_{MR}$ is expected to decay inversely with the number of axons within a field of view. However, for rodent MRI, in which the MRI voxels have the same surface area as the histological patches, the precision in the estimation of the effective radius is similar for both modalities.

Overall, dMRI provides a precise measurement of the largest axons, which are captured within an MRI voxel. In contrast, histology, so far, mainly probes the bulk that consists of smaller axons with high precision. Therefore, both modalities are complementary, especially in human MRI for which the voxels are significantly larger than a typical histological sample.

## Measuring the bulk of the axon distribution using MRI

As the signal attenuation inside axons, *Equations (5)-(6)*, scales as $\ln S \sim - g^2 r_{\text{eff}}^4$, getting to two-times smaller $r_{\text{eff}}$ would require another four-fold increase in gradients. However, even with stronger gradient systems, the main bottleneck might be the missing prior knowledge about the shape of the expected axon radius distribution $h(r)$. Even when assuming a particular functional form of $h(r)$, one is limited to estimating a *single parameter* to describe the axon radius distribution, whereas realistic distributions such as the generalized extreme value distribution (*Sepehrband et al., 2016*) are parameterized by at least two variables. Hence, the reconstruction of $h(r)$ from only $r_{\text{eff}}$ is technically ill-posed and, as such, prone to mis– or over–interpretation due to biases towards user-defined distribution shapes and parameters, even more so if confounding factors such as dispersion or fixed diffusivities are ignored (*Assaf et al., 2008*; *Barazany et al., 2009*; *Horowitz et al., 2015*; *McNab et al., 2013*).

With unknown $h(r)$, only a single metric representing the entire distribution, that is $r_{\text{eff}}$, for which the strength of tail-weighting is determined by the gradient pulse width, can be estimated reliably. In the best case, that is the narrow-pulse limit $\delta \ll t_c$, see 'From $D_a^\perp$ to effective MR radius', $r_{\text{eff}}$ will depend on the fourth rather than the sixth order moment of $h(r)$, that is $r_{\text{eff}} \equiv \sqrt{\langle r^4 \rangle / \langle r^2 \rangle}$ (*Burcaw et al., 2015*), thereby reducing, but not eliminating the difference between $r_{\text{eff}}$ and $\bar{r}$. Other methods, such as oscillating gradient diffusion weighting or double diffusion encoding, may provide other sources of contrast encoding different aspects of the size distribution (*Jiang et al., 2016*), although the low-frequency limit of the oscillating-gradient attenuation has been shown to be equivalent to the Neuman's limit, not providing any extra information (*Novikov et al., 2019*). It can be hypothesized that the combination of methods could perhaps recover more accurate information on the underlying $h(r)$.

## Dot fraction

The presence of isotropic immobile water $f_{\text{im}}$ has been conjectured by *Stanisz et al. (1997)* as water possibly restricted inside the soma of various cell types, such as neurons or oligodendrocytes. Several previous studies, for example *Veraart et al. (2019)*, *Tax et al. (2019)*, and *Dhital et al. (2019)*, demonstrated with various diffusion encoding strategies that in vivo dMRI is practically insensitive, that is < 0.2%, to such signal contributions in the healthy white matter of the living human brain, excluding the cerebellum (*Tax et al., 2019*). The lack of sensitivity of dMRI to immobile water might be explained by a small volume fraction, a short $T_2$ relaxation time, and/or a fast water exchange rate on the scale of our diffusion time $\Delta = 30\,\text{ms}$ for treating them as coming from separate compartments. In contrast, the dot compartment has been observed in fixed brain samples in various studies (*Stanisz, 1997*; *Alexander et al., 2010*). The origin of this signal contribution is not well understood yet, but the still water compartment needs to be considered when validating or studying biophysical models in fixed tissue.

In this work, for the human experiments, we build upon the previously reported observations to fix $f_{\text{im}} = 0$ in the healthy white matter to avoid fitting degeneracies that are associated with the poor conditioning of model (iii). However, any underestimation of the dot compartment, for example due to fixing $f_{\text{im}} = 0$, leads directly to an underestimation of the effective MR radius, see *Figure 1*.

Therefore, in future studies, especially those that focus on the developing, aging, or pathological brain, we encourage the independent measurement of the dot compartment to complement the axon radius acquisitions. The fast measurement of the dot compartment is promoted by the availability of spherical diffusion-encoding, as demonstrated by *Dhital et al. (2019)*, and *Tax et al. (2019)*.

In our ex vivo experiments, the measurement of the dot compartment is based on the diffusion-weighing in the direction parallel to the average fiber direction in the CC at the maximal $b$-value of 100 ms/µm². The measured signal fraction of such a still water compartment in our fixed brain samples was in the range of 8 to 17%, in line with *Stanisz et al. (1997)*. Applying a radial or planar diffusion-weighting filter prior to this measurement would suppress any contribution of anisotropic signal compartments, such as crossing or dispersed axons, to the isotropically restricted dot compartment. Although we aimed to minimize this confounding factor by using a very high $b$-value (*Dhital et al., 2019*), the dot compartment fraction, and as such the effective MR radius, might be slightly overestimated because of various complex fiber configurations. Additional confounding factors are listed in the following section.

## Confounding factors

The apparent discrepancy between histology and dMRI, when confounding factors such as extra-axonal water (*Burcaw et al., 2015*; *Fieremans et al., 2016*; *Lee et al., 2018*) and orientational dispersion (*Drobnjak et al., 2016*; *Nilsson et al., 2012*) are addressed, is mainly a result of the difference between $\bar{r}$ and $r_{\mathrm{eff}}$ – that is between the bulk and the tail of axonal distribution. This already provides an important insight into the discussion on why the radii reported in the literature vary so much between the methods. When comparing apples-to-apples, despite the excellent agreement observed in this study between $r_{\mathrm{MR}}$ and its histological counterpart $r_{\mathrm{eff}}$, in our own histological validation we still observed a small, yet consistent overestimation of between 5 and 20% in axon radius $r_{\mathrm{MR}}$ using dMRI. Aside from the previously discussed dot compartment, various other factors might contribute to this discrepancy.

First, an underlying assumption of all studies targeting the measurement of the axon radius is specificity: that the signal observed at these high $b$-values could be attributed exclusively to the intra-axonal space. However, this assumption is not established nor in our opinion is it justified given that water resides in all cellular compartments of the central nervous system. We cannot exclude that water trapped in other 'stick'-like features such as the radiating processes of astrocytes contribute observable signals; it has been previously reported, but also observed in our histological sample, that such glial processes can have large diameters, up to 7 µm in our sample. In the future, this contribution could be investigated using the increased cellular specificity of (diffusion-weighted) spectroscopy (*Palombo et al., 2016*; *Shemesh et al., 2017*; *Ligneul et al., 2019*).

Second, *Stanisz et al. (1997)* and, more recently, *Palombo et al. (2019)* demonstrated that at shorter diffusion times, the signal contribution from cell bodies might be characterized by a specific $b$-value dependent signature (*Neuman, 1974* and *Murday and Cotts, 1968*) that might enable the extraction of MR effective cell body sizes in both the white and gray matter (*Palombo et al., 2019*). In this study, the potential $b$-value dependent signal contribution of cell bodies was unaccounted for, and, given our and other (e.g. *Sampaio-Baptista et al., 2019*) observations of a finite cell body volume fraction, the axon radius measurements could be biased. However, deviations to the power law scaling due to the presence of cell bodies are more likely to be expected in the GM because of larger volume density of large somas in comparison to the WM (*Palombo et al., 2019*). In our histological sample, we observed a significant volume fraction of cell bodies in the genu of the CC, that is 5%, but due to unknown compartmental relaxation times, the associated, yet more important, signal fraction is unknown (*Lampinen et al., 2019*). Regardless, a biophysical model parameterized by various volume fractions, axon radii, soma sizes, and compartmental diffusivities may be poorly conditioned and degenerate.

Third, along-axon undulations (*Nilsson et al., 2012*) and curvature (*Özarslan et al., 2018*) might result in an increased apparent radial diffusivity and, as such, contribute to an overestimation of the axon radius using dMRI, especially for increasingly long diffusion times (*Lee et al., 2019*; *Brabec, 2019*).

Finally, the estimation of the MR effective axon radius depends on the unknown intrinsic diffusivity $D_0$ of the axoplasm. In ex vivo samples of a well-aligned WM bundle, one could estimate $D_0$

directly by exploring the time dependence of the apparent diffusivity at very short diffusion times, (**Mitra et al., 1993**). In this study, we were not able to achieve a reproducible and precise estimate of $D_0$ and opted to use the longitudinal diffusivity $D_a^{\parallel}$ as a proxy for $D_0$, with $D_a^{\parallel} \leq D_0$. Therefore, we might actually underestimate the positive bias in the estimation of $\hat{r}_{\text{eff}} \sim (D_0)^{1/4}$ (**Equation 10**). However, the propagation of the error in the estimation of $D_0$ to $\hat{r}_{\text{eff}}$ is strongly reduced by the fourth root relation between both metrics.

## Inter-species variability

In our study, the effective MR radius in humans was significantly higher than in rats when comparing similar regions of interest, for example the midbody of the CC. This difference is in agreement with several studies that compared the callosal fiber composition as a function of the brain size of various mammals and concluded that large brains have more large axons and an increased maximal radius, whereas the bulk of axons is not altered (**Olivares et al., 2001**; **Schüz and Preiβl, 1996**; **Liewald et al., 2014**). Since the effective MR radius is predominantly sensitive to the larger axons, observed differences between humans and rats will be amplified when comparing effective MR radii. Overall, this observation favors future application of MR axon radius mapping in species with relatively large brain sizes.

## Gray matter

Although this work mainly focuses on the WM, we do report significantly different signal scaling for the cortical GM. We suggest that the proton exchange between dendrites and interstitial water might explain this scaling behavior, in particular due to the convex scaling with $b^{-1/2}$. However, the abundance of cell bodies in the gray matter might confound this analysis (**Palombo et al., 2019**). Moreover, the study of the cortical GM is challenged by its low SNR and susceptibility to partial voluming. Nonetheless, we conclude that the stick assumption does not hold in the cortical GM and that biophysical models building upon that assumption must be interpreted with caution if applied to tissue regions outside of WM.

## Conclusion

In summary, we provide a realistic perspective on MR axon radius mapping by showing MR-derived effective radii that have good quantitative agreement with histology. First, we compared the MR-derived axon radii directly to confocal microscopy of the same rat brain samples. Second, the distribution of dMRI-derived $r_{MR}$ of the living human brain falls almost entirely within the range spanned by histology-derived $r_{eff}$ that has been reported in the literature — even without introducing a putative axonal shrinkage factor. This estimation is inherently bound to a single scalar $r_{eff}$ that encodes moments of the axon distribution, which is – by virtue of the signal encoding – dominated by the largest axons. Therefore, the average axon radius $\bar{r}$ and $r_{eff}$ can be practically considered as two complementary metrics probing the underlying axon caliber distribution: histology, so far, mainly probes its bulk, that is $\bar{r}$, while dMRI probes $r_{\text{MR}} = \hat{r}_{\text{eff}}$, that is its tail. Due to the intrinsic bias of MR-derived axon radii to larger axons, clinical applications should focus on pathologies that specifically target those larger axons, until other methods are developed that probe the smaller axon diameter.

## Materials and methods

**Key resources table**

| Reagent type (species) or resource | Designation | Source or reference | Identifiers | Additional information |
| --- | --- | --- | --- | --- |
| Antibody | anti-Neurofilament 160/200 (Mouse monoclonal) | Sigma Aldrich | Cat# N2912 (clone RMdo20) | 2.5 µg/mL |
| Antibody | anti-GFAP (rabbit polyclonal) | Thermo Fisher Scientific | Cat# PA1-10019 | 1:1000 |
| software. algorithm | ImageJ | imagej.nih.gov/ij/ | RRID: SCR003070 | 1.52q |

*Continued on next page*

*Continued*

| Reagent type (species) or resource | Designation | Source or reference | Identifiers | Additional information |
|---|---|---|---|---|
| software. algorithm | FSL | fsl.fmrib.ox.ac.uk/fsl/ | RRID: SCR002823 | v6 |
| software. algorithm | MRtrix | www.mrtrix.org | RRID: SCR006971 | v3.0 |
| software. algorithm | FreeSurfer | surfer.nmr.mgh. harvard.edu | RRID: SCR001847 | v6.0.0 |
| Other | DAPI | Sigma Aldrich | Cat# D9542 | 500nM |

## MRI of fixed rat brain tissue

### Ethics

Animals used in this study were handled in agreement with the European FELASA guide-lines and all procedures were approved by the Champalimaud Animal Welfare Body and by the national authorities, Direção Geral de Alimentação e Veterinária, Lisbon, Portugal, under the approved protocol number 0421/000/000/2016. All animal care procedures were conducted in agreement with the European Directive 2010/63, at the vivarium of the Champalimaud Foundation, a research facility part of CONGENTO, project number Lisboa-01–0145-FEDER-022170.

### Sample preparation

Three Long Evans rats (Female, 12-weeks-old) were transcardially perfused using 4% paraformaldehyde. The extracted brains were kept for 24 hr in 4% paraformaldehyde and washed using PBS over two days (changed daily). Given our focus on the CC of the rat brains, we will refer to the samples as CC#1, CC#2, and CC#3.

### MRI scanning

i. Multi-shell dMRI data: The three samples were scanned on an 16.4T MR scanner (Bruker Bio-Spin) at room temperature with $\Delta/\delta = 20/7.1\,\text{ms}$ interfaced with an AVANCE IIIHD console and a micro2.5 imaging probe with maximal gradient amplitude $G_{\max} = 1500\,\text{mT/m}$. Diffusion-weighting was applied using a RARE sequence in the midsagittal plane along 60 gradient directions for a densely sampled spectrum of 18 different $b$-values up to 100 ms/μm². Furthermore, $\text{TR}/T_{\text{E}} = 2400/30.4\,\text{ms}$ and the spatial resolution was $100 \times 100 \times 850\,\text{μm}^3$.

ii. 'dot fraction' $f_{\text{im}}$: we acquired 60 repeated measurements of diffusion-weighing applied in the direction parallel to the average fiber direction in the corpus callosum (CC) at the maximal $b$-value of 100 ms/μm². The average fiber orientation was defined as the first eigenvector of the dyadic tensor (*Jones, 2003*) that was computed from the voxelwise first eigenvectors of the diffusion tensors that were estimated by fitting the DTI model to the lowest $b$-values, that is $b<5\,\text{ms/μm}^2$, of the multi-shell data in each voxel within the manually segmented CC (*Basser et al., 1994*).

The average SNR for $S|_{b=0}$ was 195 and some examples of the acquired images at various low and high $b$-values are shown in *Figure 10*, where the quality of the raw data can be evaluated. Notably, since images are spherically averaged over many directions, the signal is characterized by high SNR even at high $b$-values.

### Data analysis

From the multi-shell data, the spherically-averaged signals $\overline{S}(b)$ are estimated per $b$-value as the zero$^{\text{th}}$ order spherical harmonic using a Rician maximum likelihood estimator of the even order spherical harmonic coefficients up to the 6$^{\text{th}}$ order (*Sijbers et al., 1998*).

The spatially localized dot fraction $f_{\text{im}}$ is computed as the signal estimated from the repeated ($N = 60$) measurements in the direction parallel to the principal fibre direction using a Rician maximum likelihood estimator with pre-computed noise level (*Veraart et al., 2016*), normalized by the respective non-diffusion weighted signal. We compute the *dot*-free signal in each voxel as follows

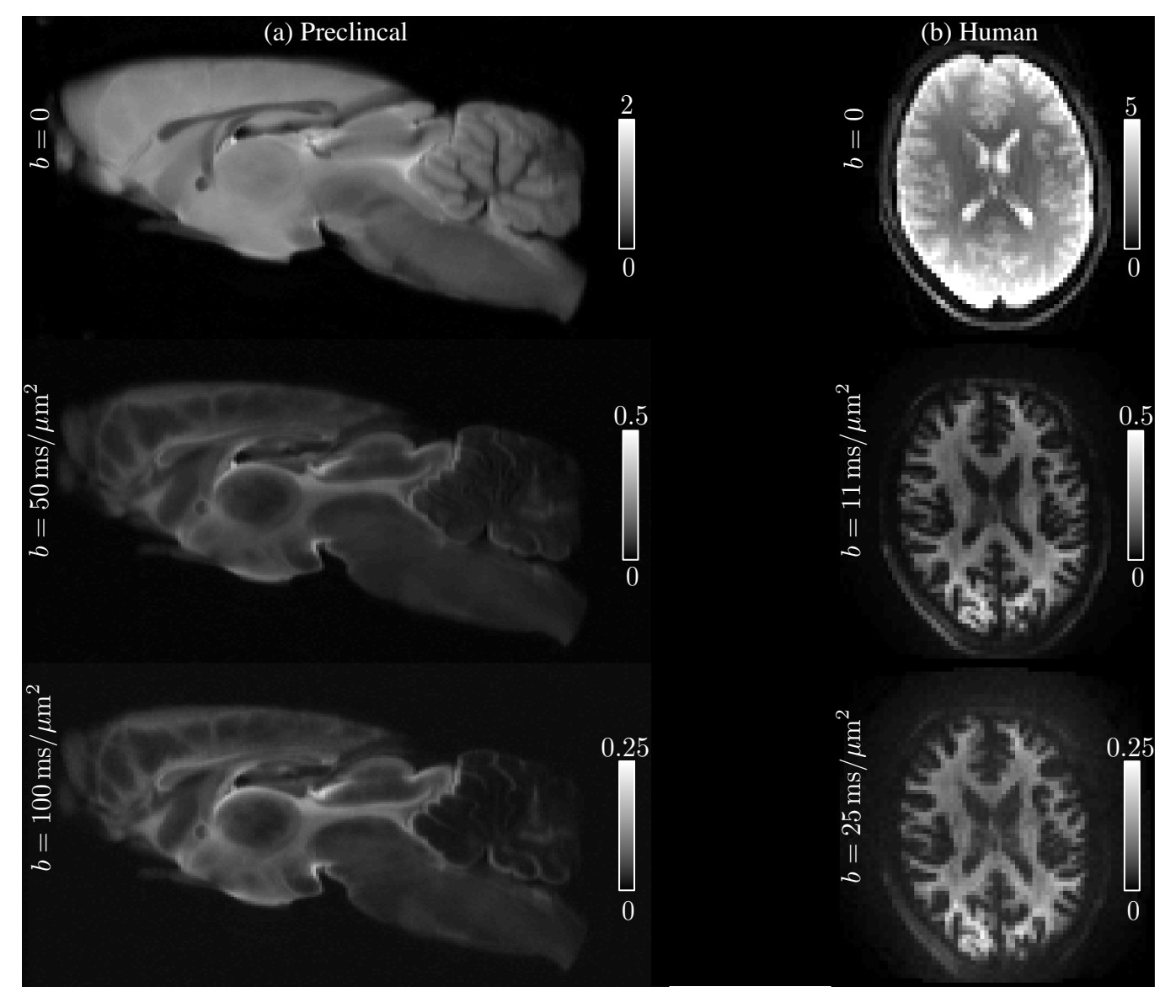

**Figure 10.** Raw data. The spherically-averaged diffusion-weighted images, prior to any other image corrections, are shown for various low and high *b*-values for one rat brain sample (**a**) and one human subject (**b**).

$\overline{S}_*(b) = \overline{S}(b) - f_{\mathrm{im}}$. In the remainder of the work, analyses were done on both 'dot contaminated' $\overline{S}(b)$ and 'dot free' $\overline{S}_*(b)$ signals.

The intra-axonal radial diffusivity $\hat{D}_a^{\perp}$ is estimated voxelwise by fitting **Equation 2** with $f_{\mathrm{im}} = 0$ to $\overline{S}^*(b \geq 20)$ using a nonlinear least squares estimator (code is available for download on GitHub [**Veraart and Novikov, 2019**; https://github.com/NYU-DiffusionMRI/AxonRadiusMapping; copy archived at https://github.com/elifesciences-publications/AxonRadiusMapping]).

Thereafter, the estimated effective axon radius $r_{\mathrm{MR}}$ is derived from $\hat{D}_a^{\perp}$ using **Equation 6**. The alternative approach, that is the simultaneous estimation of $\hat{D}_a^{\perp}$ and $f_{\mathrm{im}}$ from $\overline{S}(b \geq 20)$ is very poorly conditioned. Hence, disentangling both parameters from only the linearly-encoded multi-shell data is impossible, even at unrealistically high SNR.

## Histology of fixed rat brain tissue

Full details of the immunohistochemistry for sample preparation, confocal microscopy, and image analysis are provided in *Nunes et al. (2017)*. Study-specific elements are described below.

### Sample preparation

After MRI scanning, free-floating horizontal sections 50 $\mu$m-thick were collected from the medial lateral center of two rat brains, CC#1 and CC#2, corresponding to the MR imaged volume. For CC#2, we used antibodies against neurofilaments 160/200 (axonal marker; Sigma-Aldrich, cat.# N2912) and GFAP (astrocytes marker; ThermoFisher Scientific, cat.# PA1-10019), as well as a staining for cell bodies using DAPI (Sigma-Adrich, cat.# D9542). For CC#1, the staining was limited to the neurofilaments to focus on the axon radius count.

### Confocal microscopy

A Zeiss LSM 710 laser scanning confocal microscope was used for immunohistochemistry image acquisition. A tile scan using a 10× objective (EC Plan Neofluar, numerical aperture = 0.3, Zeiss, Germany) was used to cover the entire CC. Various ROIs were imaged using a 63× immersion objective (Plan Apochromat, numerical aperture = 1.4, Zeiss, Germany) in confocal mode, with pixel resolution of $65 \times 65 \times 150$ nm$^3$ and field-of-view of $135 \times 135$ $\mu$m$^2$ (*Figure 11*). The placement of the ROIs is shown in *Figures 4* and *5* for CC#1. and CC#2, respectively.

### Confocal microscopy data analysis

Images were processed using the ImageJ software. Noise suppression of the confocal single frames was done using a subsequent application of a 2D anisotropic diffusion filter and bandpass filtering in the frequency domain, (*Nunes et al., 2017*). Thereafter, axons were identified as particles with a minimum area size of 0.2 $\mu$m$^2$ and a circularity larger than 0.4 in the confocal images that were stained for neurofilaments. The number of extracted axons varied from about 500 to 2000, depending on the placement of the ROI. The long axes of fitted ellipsoids served as proxies for the respective axon radii. For each ROI, we obtain a distribution of axon radii $h(r)$ from which we compute the associate effective radius $r_{\mathrm{eff}}$ using *Equation 9*.

## In vivo MRI of human brain

### Ethics

Data were acquired after obtaining written informed consent and consent to publish. The project was approved by the Cardiff University School of Psychology Ethics Committee (approval number EC.06.05.02.891).

### Subjects

Four healthy volunteers (3 males and 1 female between 22 and 45 years) were recruited for this study. We will refer to the human subjects as 22M, 25F, 32M, and 45M to encode both the age and gender. The data were collected under the approval of the Cardiff University School of Psychology Ethics Committee.

### MRI scanning

All four subjects were scanned on a Siemens Connectom 3T MR scanner using a 32-channel receiver coil. The 300 mT/m gradient system was used to achieve $b$-values up to $25\,\mathrm{ms}/\mu\mathrm{m}^2$. The diffusion gradients were characterized by $\Delta/\delta = 30/13\,\mathrm{ms}$ and maximal gradient amplitude of 289 mT/m. Diffusion weighting was applied along 60 isotropically distributed gradient directions (*Jones et al., 1999*) for $b = 1, 3, 5, 7, 9, 11, 12.1, 13.5, 15, 16.9, 19.1, 21.7,$ and $25\,\mathrm{ms}/\mu\mathrm{m}^2$, with TR/T$_\mathrm{E}$ : 3500/62 ms, matrix: $74 \times 74$, and 42 slices with a spatial resolution of $3 \times 3 \times 3\,\mathrm{mm}^3$. The average SNR for $S|_{b=0}$ was 52. See *Figure 11* for the image quality and contrast at various $b$-values.

The dot compartment was not measured directly (see *Dhital et al., 2018* and *Tax et al., 2019*).

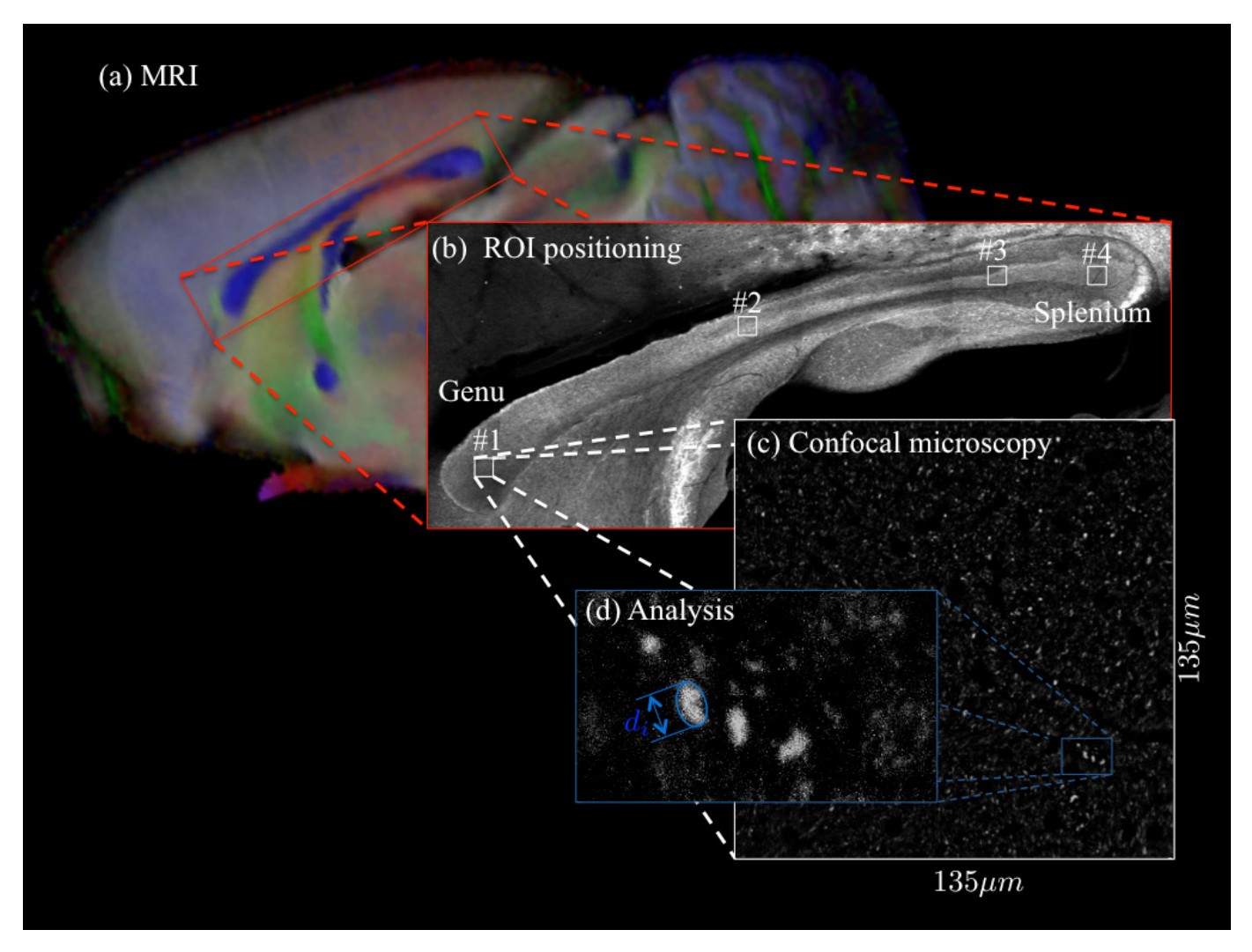

**Figure 11.** For two brain samples, MR scanning (**a**, color encoded FA map) was followed by low (**b**) and high (**c**) resolution confocal microscopy with staining for neurofilaments to identify the axons. The low-resolution image was used to position various ROIs, whereas the axon caliber distributions were extracted from the high-resolution image of the corresponding ROIs. The long axes of fitted ellipsoids served as proxies for the respective axon diameters (**d**).

## Data analysis

Image processing was done according to the DESIGNER pipeline (*Ades-Aron et al., 2018*) using the FSL (*Smith et al., 2004*) and MRtrix (*Tournier et al., 2019*) software packages. In particular, MPPCA noise estimation and denoising (*Veraart et al., 2016*) were used for estimating noise maps $\sigma(x)$ by exploiting the inherent redundancy in dMRI data. The positive signal bias, inherent to low-SNR magnitude MR data, was removed by using the method of moments (*Koay and Basser, 2006*), where the denoised signal was used as a proxy for the Rician expectation value. Denoised and Rice-floor-corrected images were subsequently corrected for Gibbs ringing (*Kellner et al., 2016*), geometric eddy current distortions and subject motion (*Andersson and Sotiropoulos, 2016*). The pipeline is available on https://github.com/NYU-DiffusionMRI/DESIGNER (*Ades-Aron and Veraart, 2018*). We used tract-density imaging (*Calamante et al., 2010*) based on whole-brain probabilistic fiber-tracking (*Tournier et al., 2019*) of the $b = 5\,\mathrm{ms}/\mu\mathrm{m}^2$-shell for identifying all WM voxels. To avoid voxels affected by partial voluming with the gray matter (GM), an additional, more conservative, segmentation was obtained by omitting all voxels with a fractional anisotropy smaller than 0.6. In addition, the cortical GM was segmented using FreeSurfer (*Dale et al., 1999*).

## Acknowledgements

All authors would like to thank Prof. Mark D Does (Vanderbilt University) for the remmiRARE sequence used in this study, supported by grant number NIH EB019980, and Dr. Erika Raven for discussions and comments on the manuscript. JV is a Postdoctoral Fellow of the Research Foundation - Flanders (FWO; grant number 12S1615N). DN and NS are supported by the European Research Council (ERC) under the European Union's Horizon 2020 research and innovation programme (Starting Grant, agreement No. 679058). EF and DSN were supported by the NIH/NINDS award R01NS088040 and research was performed as part of the Center of Advanced Imaging Innovation and Research (CAI2R, www.cai2r.net), an NIBIB Biomedical Technology Resource Center (NIH P41 EB017183). The Connectom data were acquired at the UK National Facility for in vivo MR Imaging of Human Tissue Microstructure funded by the EPSRC (grant EP/M029778/1), and The Wolfson Foundation. DKJ is supported by a Wellcome Trust Investigator Award (096646/Z/11/Z) and a Wellcome Trust Strategic Award (104943/Z/14/Z).

## Additional information

### Funding

| Funder | Grant reference number | Author |
|---|---|---|
| Fonds Wetenschappelijk Onderzoek | 12S1615N | Jelle Veraart |
| National Institute of Neurological Disorders and Stroke | R01NS088040 | Els Fieremans<br>Dmitry S Novikov |
| National Institute of Biomedical Imaging and Bioengineering | P41 EB017183 | Els Fieremans<br>Dmitry S Novikov |
| H2020 European Research Council | 679058 | Noam Shemesh<br>Daniel Nunes |
| Engineering and Physical Sciences Research Council | EP/M029778/1 | Derek K Jones |
| Wellcome | 096646/Z/11/Z | Derek K Jones |
| Wellcome | 104943/Z/14/Z | Derek K Jones |

The funders had no role in study design, data collection and interpretation, or the decision to submit the work for publication.

### Author contributions

Jelle Veraart, Conceptualization, Data curation, Software, Formal analysis, Funding acquisition, Validation, Investigation, Visualization, Methodology, Writing - original draft, Writing - review and editing; Daniel Nunes, Conceptualization, Data curation, Software, Formal analysis, Visualization, Methodology, Writing - original draft, Writing - review and editing; Umesh Rudrapatna, Data curation, Writing - original draft, Writing - review and editing; Els Fieremans, Conceptualization, Funding acquisition, Writing - original draft, Writing - review and editing; Derek K Jones, Resources, Data curation, Funding acquisition, Writing - original draft, Writing - review and editing; Dmitry S Novikov, Conceptualization, Formal analysis, Supervision, Funding acquisition, Validation, Investigation, Visualization, Methodology, Writing - original draft, Writing - review and editing; Noam Shemesh, Conceptualization, Resources, Data curation, Supervision, Visualization, Methodology, Writing - original draft, Writing - review and editing

### Author ORCIDs

Jelle Veraart https://orcid.org/0000-0003-0781-0420
Daniel Nunes https://orcid.org/0000-0001-8882-3228
Els Fieremans https://orcid.org/0000-0002-1384-8591
Dmitry S Novikov https://orcid.org/0000-0002-4213-3050
Noam Shemesh https://orcid.org/0000-0001-6681-5876

## Ethics

Human subjects: Human studies were carried out under a protocol (EC.06.05.02.891) approved by Cardiff University School of Psychology Ethics Committee. Written informed consent and consent to publish was obtained from all participants.

Animal experimentation: Animals used in this study were handled in agreement with the European FELASA guide-lines and all procedures were approved by the Champalimaud Animal Welfare Body and by the national authorities, Direção Geral de Alimentação e Veterinária, Lisbon, Portugal, under the approved protocol number 0421/000/000/2016. All animal care procedures were conducted in agreement with the European Directive 2010/63, at the vivarium of the Champalimaud Foundation, a research facility part of CONGENTO, project number Lisboa-01-0145-FEDER-022170.

## Decision letter and Author response

Decision letter https://doi.org/10.7554/eLife.49855.sa1
Author response https://doi.org/10.7554/eLife.49855.sa2

# Additional files

## Data availability

All source data files generated or analysed during this study have been deposited in Dryad Digital Repository (http://doi.org/10.5061/dryad.4qrfj6q66).

The following dataset was generated:

| Author(s) | Year | Dataset title | Dataset URL | Database and Identifier |
|---|---|---|---|---|
| Veraart J, Nunes D, Rudrapatna U, Fieremans E, Jones DK, Novikov DS, Shemesh N | 2019 | Data from: Noninvasive quantication of axon radii using diffusion MRI | http://doi.org/10.5061/dryad.4qrfj6q66 | Dryad Digital Repository, 10.5061/dryad.4qrfj6q66 |

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
