## [Decision Letter]

**Acceptance summary:**

This study focuses on a longstanding and important question in the field of diffusion MRI, namely the measurement of axon diameters. Importantly, until now, the accurate estimation of axon diameter mapping with non-invasive techniques such as diffusion MRI has been elusive due to a lack of sensitivity in the signal. The authors provide compelling evidence using sophisticated modeling that axon diameters can be estimated for largest axons when eliminating confounding factors such as extra-axonal water and axonal orientation dispersion. Data of fixed rat brains and optical microscopy of the same specimen are presented showing good quantitative agreement for MR-derived axon diameters. Finally, in vivo data from Connectom 3T scanning is presented which shows the feasibility of mapping axon diameters in healthy subjects. The work is therefore of interest to a broad scientific audience ranging from physicists to cognitive neuroscientists.

**Decision letter after peer review:**

Thank you for submitting your article "Noninvasive quantification of axon diameter using diffusion MRI' for consideration by *eLife*. Your article has been reviewed by three peer reviewers, including Birte Forstmann as the Reviewing Editor and Reviewer #1, and the evaluation has been overseen Floris de Lange as the Senior Editor.

The reviewers have discussed the reviews with one another and the Reviewing Editor has drafted this decision to help you prepare a revised submission.

Summary:

Overall, this manuscript is well written, interesting, timely and will help resolve the debate in the field. We therefore suggest revising the manuscript to address the points raised by the reviewers which are outline below.

Reviewer 1:

This paper introduces an interesting and surprisingly simple method for estimating axon diameter (potentially in vivo). Their approach relies on three key ingredients:

– ensure that the diffusion sensitisation (b value) is high enough to eliminate extra-axonal water

– use powder averaging across diffusion gradient orientations to eliminate confounds due to orientation of axons

– estimate intra-axonal perpendicular diffusivity from a simple power-law formula predicted by the theory

The intra-axonal perpendicular diffusivity is then used to estimate an "effective axonal radius" which describes the tail of the axon radius distribution under certain assumptions.

In general, this is a potentially important contribution. I have the following comments which I am sure the authors will be able to address:

1) Presentation

I found the presentation of the theory unnecessarily dense and difficult to follow. A simple diagram might help. Something like Figure 1 but with a single line showing the prediction from a model with f_im_ as well as D_perp_>0 (so with non-zero intercept of the nonlinear part and a negative intercept of the linear part). It would also be helpful to indicate on the same diagram that the slope on the right-hand side depends on D_para_, and that the intercept is f_im_, but the 'dashed line' intercept is a function of f_im_ and D_perp_ (and D_para_?). The x-axis could be double-labelled with both 1/b and b. And also it would help to have vertical sections in the graph indicating the b value regimes (e.g. clinically-feasible, vs. Connectom vs. small-bore scanners, vs. low-b regime).

2) Modelling

Having played with the models (i)-(viii) an little, I see lots of degeneracy between f_im_ and D_perp_ over a wide range of parameter values which is not surprising: unless some curvature is visible in the data (above the noise level), it is difficult to disentangle the contributions of f_im_ and D_perp_ to the negative intercept. In the data that is shown (e.g. Figure 5), the points fall in a straight line and so there is no curvature to help disentangle f_im_ and D_perp_. The authors assumed that f_im_=0 for the in vivo data, but is this really justified (could there not still be very slow diffusing water that is unmodelled) and what about ex vivo?

In general, some sort of analysis of when the degeneracy breaks down (as a function of the max b value attainable and the other params like D_para_) would be helpful here. For example looking at the full posterior distribution and not just point estimates of the parameters (I don't find AIC values very helpful compared to looking at the full posterior distribution).

3) Axon diameter estimation

It would be helpful if the authors could unpack how they get to r_eff_ as a function of D_perp_. I can see that the diffusion in a cylinder formula gives rise to a r^4^ dependence of the log(S) in the regime that the acquisition are made in. But then to go from there to r_eff_ (which is the ratio of 6th to 2nd moment) is a stretch for me. Is it simply by doing a Taylor expansion of S=exp(-a*r^4^) around zero inside the integral in Equation 7? If that is the case then perhaps an appendix would not hurt. Also, it is not clear how accurate it is to use the Taylor expansion.

Also on axon diameters, the authors make it quite clear that they don't like methods that make explicit distributional assumptions on the axon diameter (e.g. AxCaliber) – but I think it would be interesting to compare them with the author's approach. Looking at the histology data that is presented, one wonders how accurate a gamma distribution would be. With a distributional assumption there would be no need for the Taylor expansion above, and everything can be done keeping the exponentials and directly inverting the equations to get the parameters of the gamma distribution. How does this compare to the r_eff_ proposed by the authors?

4) Are the results biologically sound?

Generally, I found that there was not enough in terms of showing results that indicate the technique actually works well, e.g. in Figure 8: is there a way to show a similar map from histology?. Or in general show that inter-areas variation in radius from histo correlate with inter-area variation from MR in the ex vivo data. The only comparison between the two modalities is done in Figure 7 in a single region.

Similarly for the in vivo data: Figure 10: is there any evidence that the intervoxel variation is meaningful? There are zones of reduced radius in the lateral frontal lobes near the cortex – are those meaningful?

5) Exchange

The authors present an interesting extra source of information, in that an exchange model makes a different prediction at high b values and they found evidence for exchange in GM. Can you convince the reader that this is not just partial volume effects (e.g. multiple pools of water with different diffusion coefficients and no exchange?) would that for example induce a curve with the opposite convexity? As GM is likely to have more partial volume issues I think this is a valid question that needs addressing.

Also on exchange: My understanding of the Karger model is that it assumes that exchange happens in situ (a molecule would change its behaviour from e.g. slow to fast diffusion with some probability instantaneously). But in reality exchange happens at the membrane. Does that invalidate the equations? Can the equations be derived here in an appendix?

6) Presentation of the data

Single voxel data is never shown and so it is difficult to tell how noisy the signal vs. 1/sqrt(b) curves actually are.

7) Data sharing

The authors are to be commended on sharing their data. However the way they have done it is not optimal in that they only provide raw data with no particular documentation or curation. The shared data set would strongly benefit if you would add the following:

– include preprocessed data not just raw data (including the D_perp_ and r_eff_ maps) – or at least provide code to generate the maps and do the preprocessing

– match data format between human and rodent

– include documentation

– avoid lsm format as it is proprietary – maybe use tiff instead?

– include processed histo data?

Reviewer 2:

This study focuses on a longstanding and important question in the field of diffusion MRI, namely the measurement of axon diameters. Importantly, until now, the accurate estimation of axon diameter mapping with non-invasive techniques such as diffusion MRI has been elusive due to a lack of sensitivity in the signal. The authors provide compelling evidence using sophisticated modeling that axon diameters can be estimated for largest axons when eliminating confounding factors such as extra-axonal water and axonal orientation dispersion. Data of fixed rat brains and optical microscopy of the same specimen are presented showing good quantitative agreement for MR-derived axon diameters. Finally, in vivo data from Connectom 3T scanning is presented which shows the feasibility of mapping axon diameters in healthy subjects.

Major comments:

Generalizability of the data:

My main concern is that the MRI-based axon diameter modeling was only evaluated in the corpus callosum. It would be important to see whether the modeling also holds in other fiber tracts, e.g., fronto-occipital fasciculus.

This is something that the authors should ideally address, but in case this is not feasible, at least comment on.

Reviewer 3:

This is an impressive work combining well-thought out theory with experimental data only recently available, particularly for the human studies, using the Connectom system to provide gradient strengths some 4 times larger than available on commercial scanners. The mix of pre-clinical data with rat CC for which histological distributions of axon diameters was measured, with human data (using somewhat less gradient strength than available on the animal system) and only literature histology is justified and actually adds strength to the comparison of experimental with theoretical considerations. The lack of any attempt to measure the "dot" component in humans is less justifiable in my view though that might have significantly added to the scan time and further comments on this might be appreciated. The authors recognize the limitations of their assessment in having to rely upon a rather "weighted" version of the distribution which gives an output index well into the tail of the distribution, the larger axons, but at least the measures are getting closer to the actual size of the median axon values than those reported in the past with more standard gradient strengths and perhaps dubious modeling. It also would be helpful to perhaps add to Figure 1 or another figure the curves that would be anticipated theoretically from the exchange model of Equation 4 at such high b values, emphasizing the difference between concave and convex theoretical curves that the authors, I assume, deem to eliminate the latter model given the experimental data. Finally as a major point, in the Data Analysis section the authors explain r_*e*ff_ or *r*_MR_ from the data but this description is difficult to follow. For example, in Equation 2, how are the O(b^2^) taken into account if they are. Then, assume we now have Da(perpendicular) how does one use that with Equations 5 and 9 to get r_eff_. This must be clarified. People should be able to replicate this calculation from what is in this text.

---

## [Author Response]

Reviewer 1:[…]In general, this is a potentially important contribution. I have the following comments which I am sure the authors will be able to address:1) PresentationI found the presentation of the theory unnecessarily dense and difficult to follow. A simple diagram might help. Something like Figure 1 but with a single line showing the prediction from a model with f_im_ as well as D_perp_>0 (so with non-zero intercept of the nonlinear part and a negative intercept of the linear part). It would also be helpful to indicate on the same diagram that the slope on the right-hand side depends on D_para_, and that the intercept is f_im_, but the 'dashed line' intercept is a function of f_im_ and D_perp_ (and D_para_?). The x-axis could be double-labelled with both 1/sqrt(b) and b. And also it would help to have vertical sections in the graph indicating the b value regimes (e.g. clinically-feasible, vs. Connectom vs. small-bore scanners, vs. low-b regime).

The theoretical sections, mainly “Breaking the power law” has been revised to improve readability. Following the suggestion of the reviewer, we amended Figure 1 to visualize the difference between *f_im_*and the intercept. While *f_im_*is a signal fraction of full restricted isotropic diffusion, the intercept is a convoluted metric that combines *f_im_*with a b value dependent offset that relates to the sensitivity of MR to the radius.

2) ModellingHaving played with the models (i)-(viii) an little, I see lots of degeneracy between f_im_ and D_perp_ over a wide range of parameter values which is not surprising: unless some curvature is visible in the data (above the noise level), it is difficult to disentangle the contributions of f_im_ and D_perp_ to the negative intercept. In the data that is shown (e.g. Figure 5), the points fall in a straight line and so there is no curvature to help disentangle f_im_ and D_perp_. The authors assumed that f_im_=0 for the in vivo data, but is this really justified (could there not still be very slow diffusing water that is unmodelled) and what about ex vivo?In general, some sort of analysis of when the degeneracy breaks down (as a function of the max b value attainable and the other params like D_para_) would be helpful here. For example looking at the full posterior distribution and not just point estimates of the parameters (I don't find AIC values very helpful compared to looking at the full posterior distribution).

The degeneracy is an intrinsic limitation of multi-compartmental modeling of diffusion MRI data. Various strategies to resolve this problem have been attempted during the past decade, ranging from imposing constraints or priors, to complement the data with orthogonal measurements. The former strategy has been contested because of the potential biases that might arise from inaccurate priors, whereas the exploring novel acquisition strategies to resolve the degeneracies is currently widely studied. A promising avenue is complementing the classical Stjeskal-Tanner diffusion-weighting – as used in this study – with planar and spherical diffusion-weighted strategies. Whereas linear encoding is best-suited to probe elongated cellular structures, such as axons, the spherical encoding is most sensitive to spherical objects such as cell bodies. Tax et al., 2019, and Dhital et al., 2018, used this spherical encoding to quantify the signal fraction *f_im_*in healthy white matter. In previous work (Veraart et al., 2019) we made a similar observation, i.e. f_im_<0.2%, using linear diffusion encoding at high *b* when only considering the parallel diffusion directions. Here, we adopt the conclusive result of *f_im_*not being significant for the human data in the major white matter structures.

Tissue fixation and possibly temperature-induced alterations of the microstructure has resulted in a non-zero *f_im_*in ex vivo studies. To avoid the need for the simultaneous estimation of *f_im_*and *D_perp_,* we included a dedicated experiment to quantify *f_im_*prior to the axon diameter mapping. We recommend a similar strategy when studying clinical cohorts in future studies.

The optimization landscape of Equation 2 (shown in updated Figure 1) reveals that disentangling *f_im_*and *D_perp_*is impossible, even at unrealistically high SNR. Indeed, the contrast in error function along a valley that runs through the landscape is minimal in comparison to the noise floor, see Author response image 1. In the revised manuscript, we highlight the intrinsic degeneracy of axon diameter mapping with unknown dot compartment explicitly. We dedicate a more extensive section to the dot compartment in which we discuss the degeneracy, overinterpretation of AIC analysis, and need for measurement of the dot compartment in atypical cohorts.

**Author response image 1. respfig1:** The optimization landscape of model (III) shows a shallow valley, relative to the noise floor, for a simulation that mimics the human component of the study. (left) The valley is shown in a 2D projection of the landscape (plot shown as a function of radius instead of D_a_^perp^). (right) The fit objective function along the valley is shown (red line) in comparison to the noise floor with an unrealistically high SNR of 250 for the non-DW signal.

3) Axon diameter estimationIt would be helpful if the authors could unpack how they get to reff as a function of Dperp. I can see that the diffusion in a cylinder formula gives rise to a r4 dependence of the log(S) in the regime that the acquisition are made in. But then to go from there to reff (which is the ratio of 6th to 2nd moment) is a stretch for me. Is it simply by doing a Taylor expansion of S=exp(-a*r4) around zero inside the integral in Equation 7? If that is the case then perhaps an appendix would not hurt. Also, it is not clear how accurate it is to use the Taylor expansion.

r_eff_ is derived from the Taylor expansion of <r^2^ exp(-a*r^4^)>/<r^2^>. We revised section “From D_perp_ to effective MR radius” to guide the reader through the mathematics behind the effective MR radius more rigorously. The accuracy of the Taylor expansion, and the model in general, for realistic settings has been addressed in a simulation section, Figure 4.

Also on axon diameters, the authors make it quite clear that they don't like methods that make explicit distributional assumptions on the axon diameter (e.g. AxCaliber) – but I think it would be interesting to compare them with the author's approach. Looking at the histology data that is presented, one wonders how accurate a Gamma distribution would be. With a distributional assumption there would be no need for the Taylor expansion above, and everything can be done keeping the exponentials and directly inverting the equations to get the parameters of the Gamma distribution. How does this compare to the reff proposed by the authors?

Sepehrband et al., 2016, published a comprehensive study on the accuracy of various parametric distribution to describe the axon distribution in the mouse corpus callosum. The generalized extreme valuedistribution consistently fitted the observed distributions better than other distribution functions, including the Gamma distribution. Most importantly, well-fitting distributions are parametrized by two or more parameters, so trying reconstructing the parametric distribution from the MR effective radius is ill-posed if these parameters are to be estimated from the same data. We simply don’t have enough information to estimate all parameters of such a well-fitting distribution. Instead of making additional assumption that will further bias the axon diameter estimates, we encourage future research directions in which the current approach is complemented with oscillating or short gradient pulse experiments to decode additional, lower-order, moments of the distribution. This might provide an avenue to reconstruct the parametric distribution based on MRI, but current hardware limits prevent us from conducting this experiment.

4) Are the results biologically sound?Generally, I found that there was not enough in terms of showing results that indicate the technique actually works well, e.g. in Figure 8: is there a way to show a similar map from histology?. Or in general show that inter-areas variation in radius from histo correlate with inter-area variation from MR in the ex vivo data. The only comparison between the two modalities is done in Figure 7 in a single region.Similarly for the in vivo data: Figure 10: is there any evidence that the intervoxel variation is meaningful? There are zones of reduced radius in the lateral frontal lobes near the cortex – are those meaningful?

The direct comparison between MRI and histology is done in 20 different patches, covering 4 different locations within the corpus callosum and two samples. For all those samples, we reported the accuracy, demonstrating good quantitative agreement, despite a small remaining overestimation. The potential confounding factors that might have contributed to this observation, yet we were unable to eliminate, are discussed in depth in the Discussion. Aside from this direct comparison, we do rely on a qualitative and indirect comparison for other brain structures, including the human corpus callosum. The trend of axon sizes within the corpus callosum, as shown in Figure 8, has been reported in various species, including the rat (Barazany et al., 2009 and Sargon et al., 2003), rhesus monkey (Lamantia et al., 1990), and human (e.g. Aboitiz et al., 1992). Although we don’t emphasize it in the manuscript, one might even argue that the slightly increased radii in the rostrum of the corpus callosum are in qualitative agreement with Sargon et al., 2003.

A thorough comparison of our MR results with the histological literature is challenged by the dependency on the axon diameter distribution to calculate the corresponding effective radius, as discussed in issue #4 above. The corpus callosum is best characterized in that respect and, as such, our validation component is limited to that commissural fiber. Please consider Author response image 2 in which we show the same trend in our human data.

**Author response image 2. respfig2:** Effective MR radius for various segments of the human CC, including rostrum (light blue) and genu to splenium (from left to right), for each of the 4 subjects. Each subject is represented by a subject-specific marker. The segmentation of the CC is shown on the right hand side.

The lack of such data in various regions across the human brain, especially in regions such as the frontal lobes, prevents us from making strong claims on meaning. Instead, we would claim that, now, we can start relying on axon diameter mapping using MRI to explore inter- and along-tract variability in the entire living brain.

5) ExchangeThe authors present an interesting extra source of information, in that an exchange model makes a different prediction at high b values and they found evidence for exchange in GM. Can you convince the reader that this is not just partial volume effects (e.g. multiple pools of water with different diffusion coefficients and no exchange?) would that for example induce a curve with the opposite convexity? As GM is likely to have more partial volume issues I think this is a valid question that needs addressing.

We don’t want to make the claim that exchange is the *only* possible explanation for the different signal decay, nor can we convince the reader that it is the most likely explanation. Palombo et el., 2019, recently demonstrated that the presence of a (non-)exchanging compartments such as somas might confound our interpretation. However, the functional form of the exchange with a “stick” compartment is novel, and we believe that bringing it up as a viable possibility has value. Especially because the convex scaling is not expected from a partial-volume contribution of a non-exchanging non-stick compartment, such as a spherical cell. By reporting the signal decay in the gray matter and posing the hypothesis of exchange, we would like to thereby trigger further research to explore different relevant biophysical processes in the gray matter, in order to develop the most adequate models of diffusion in GM. Hence, as stated in the Discussion, we acknowledge that exchange is just one out of a few different avenues to be explored and investigated.

Also on exchange: My understanding of the Karger model is that it assumes that exchange happens in situ (a molecule would change its behaviour from e.g. slow to fast diffusion with some probability instantaneously). But in reality exchange happens at the membrane. Does that invalidate the equations? Can the equations be derived here in an appendix?

As clarified by Fieremans et al., 2010, the Karger model (KM) is applicable for diffusion times *t* long enough so that the medium, coarse-grainedover the corresponding diffusion length *L*(*t*), can be viewed as one where exchange happens everywhere, as the reviewer has indeed noted.

For the intra-neurite water, this means that the exchange should be slow enough, so that it is “barrier-limited” rather than “diffusion-limited” (per Karger’s original terminology). Our estimates of the residence times exceeding 10ms ensure that this is the case, as most neurite diameters are under 1 micron and the corresponding diffusion time is therefore an order of magnitude slower. These estimates are in line with other measurements, e.g., by Yang et al., 2018, whose exchange times exceeded 100 ms. For the extra-neurite water, technically, the KM applicability requires *L*(*t*) ≫ *l_c_*,or, equivalently, *t* ≫ *t_c_*,where *l_c_*is the correlation length of the structure. Practically, the same requirements ensure that the tortuosity limit has been reached in each compartment separately, i.e., diffusion in each compartment has become Gaussian (neglecting the exchange); this Gaussian diffusion is what is assumed by the Karger model’s equations (our only difference is that we made this Gaussian diffusion anisotropic, as compared to isotropic diffusion originally considered by Karger). For our diffusion time, the diffusion length in the plane transverse to the neurites, Lt=2∙2Det≈2∙2∙130≈11μmexceeds the mean distance between them; it also exceeds the distance between beads along neurites (3-7 microns). The exact tissue parameters remain controversial while the EM segmentation and tissue quantification is underway. However, the beauty of employing very strong diffusion weighting is that the extra-neurite compartment is suppressed, therefore the above issues become irrelevant.

All in all, we believe KM may be asymptotically applicable here, in the view of suppressing the extra-neurite signal at strong *b*, as well as because the estimated exchange time a posteriorijustifies the barrier-limited exchange for the intra-neurite water. However, as KM may still be too simplistic, while giving an approximate range, we are not making strong claims about the numerical value of the exchange time in gray matter, because it is an entirely different active area of investigation, and the relevance of different biophysical effects there is still under intense debate. Our relatively short estimated exchange time (on the scale of clinically used diffusion times) may indeed explain the difference in the functional form between high- *b* signal in GM from a “stick” seen in WM, and adds value to the current debate about the need to include exchange in GM modeling.

6) Presentation of the dataSingle voxel data is never shown and so it is difficult to tell how noisy the signal vs 1/sqrt(b) curves actually are.

Please consider Author response image 3 in which we show the spherically-averaged signal decay as a function of 1/sqrt(b) for all individual voxels of the WM of one human subject (faded colored lines) and for the average across all WM voxels (blue line with markers, cf. Figure 5B). In addition, to highlight the precision, we show the signal decay in 10 arbitrarily chosen individual WM voxels. We would like to highlight that Figures 8 and 10 in the manuscript show voxel-by-voxel results that demonstrate the precision qualitatively.

**Author response image 3. respfig3:** Signal decay in a single voxel. (left) The spherically-averaged signal decay is shown as a function of 1/b for all individual voxels of the WM of one human subject (faded colored lines) and for the average across all WM voxels (blue line with markers, Figure 5B). (right) The signal decay for individual voxels is shown for 10 arbitrarily-chosen WM voxel. The dashed box is positioned the same for all graphs.

7) Data sharingThe authors are to be commended on sharing their data. However the way they have done it is not optimal in that they only provide raw data with no particular documentation or curation. The shared data set would strongly benefit if you would add the following:– include preprocessed data not just raw data (including the D_perp_ and r_eff_ maps) – or at least provide code to generate the maps and do the preprocessing– match data format between human and rodent– include documentation– avoid lsm format as it is proprietary – maybe use tiff instead?– include processed histo data?

Since there is no consensus within the neuroimaging community on how to process dMRI data, especially with a strong diffusion-weighting, we opt to share the raw data, for both MRI and microscopy, and not limit the users to the considerations and choices that we made. We would like to highlight that all image processing was done using publicly available software tools, i.e. FSL, Freesurfer, MRtrix, and ImageJ – as listed in the Key resources table. The code for the axon diameter fitting is also made available on our GitHub.

We now match the data format, NiFTi, for humans and rodents; provide the microscopic images in LSM and TIFF format, and include a more comprehensive documentation of the data in the Dryad Digital Repository.

Reviewer 2:[…]Major comments:Generalizability of the data:My main concern is that the MRI-based axon diameter modeling was only evaluated in the corpus callosum. It would be important to see whether the modeling also holds in other fiber tracts, e.g., fronto-occipital fasciculus.This is something that the authors should ideally address, but in case this is not feasible, at least comment on.

Our work focuses on theory, validation, and human feasibility. To demonstrate the human feasibility, we opted to limit ourselves to the corpus callosum because it’s the tract which has been characterized most thoroughly across multiple histological studies. Nonetheless, in Author response image 4, we show that the inter-tract variability of the effective MR axon diameter is higher than the inter-subject variability to provide on optimistic outlook on whole brain characterization in future studies. A comment is made in the revised manuscript.

**Author response image 4. respfig4:** Average effective MR radius within various tracts (color encoded as shown on the right; both hemispheres were considered simultaneously) for the 4 human subjects (encoded by marker). The line segments show the mean across the subjects. The inter-tract variability exceeds the inter-subject variability. We hypothesize that the hereby introduced technique can be used in future studies to characterize the typical, developing, or pathological brain in a wide range of species, including humans, rodents, or non-human primates.

Reviewer 3:[…]The lack of any attempt to measure the "dot" component in humans is less justifiable in my view though that might have significantly added to the scan time and further comments on this might be appreciated.

The justification of a negligible dot compartment in the healthy white matter was found in various recent publications, Dhital et al., 2018, Veraart et al., 2019, and Tax et al., 2019, which concluded independently from each other, using different techniques, that such a compartment is not significant in the major white matter pathways in the healthy adult brain. Note that the study of Tax et al. was performed on exactly the same scanner. That being said, when moving towards clinical or preclinical applications, we encourage the independent measurement of the dot compartment to complement to axon diameter acquisitions. The fast measurement of the dot compartment is promoted by the availability of spherical diffusion-encoding. We do comment more extensively on this discussion point in the revised manuscript.

It also would be helpful to perhaps add to Figure 1 or another figure the curves that would be anticipated theoretically from the exchange model of Equation 4 at such high b values, emphasizing the difference between concave and convex theoretical curves that the authors, I assume, deem to eliminate the latter model given the experimental data.

The curve is now shown in Figure 1 to showcase the significantly different signal decay.

Finally as a major point, in the Data Analysis section the authors explain r_eff_ or rMR from the data but this description is difficult to follow. For example, in Equation 2, how are the O(b^2^) taken into account if they are. Then, assume we now have Da(perpendicular) how does one use that with Equations 5 and 9 to get r_eff_. This must be clarified. People should be able to replicate this calculation from what is in this text.

This comment is in line with comments 1 and 3 from reviewer #1. We include the derivation of r_eff_ using Equations 5 to 8 explicitly. The error associated with modelling approximations, e.g. omitting O(b^2^) or the Taylor expansion in Equation 9, is estimated using a realistic simulation framework, as shown in Figure 4.